# Single-cell transcriptome analysis reveals aberrant stromal cells and heterogeneous endothelial cells in alcohol-induced osteonecrosis of the femoral head

Zheting Liao [1,2,7], Yu Jin[1,2,7], Yuhao Chu[1,2,7], Hansen Wu[2,3,7], Xiaoyu Li[1,2], Zhonghao Deng[1,2], Shuhao Feng[1,2], Nachun Chen[1,2], Ziheng Luo[1,2], Xiaoyong Zheng[4], Liangxiao Bao[1], Yongqing Xu[5], Hongbo Tan [5✉] & Liang Zhao [1,2,6✉]

Alcohol-induced osteonecrosis of the femoral head (ONFH) is a disabling disease with a high incidence and elusive pathogenesis. Here, we used single-cell RNA sequencing to explore the transcriptomic landscape of mid- and advanced-stage alcohol-induced ONFH. Cells derived from age-matched hip osteoarthritis and femoral neck fracture samples were used as control. Our bioinformatics analysis revealed the disorder of osteogenic-adipogenic differentiation of stromal cells in ONFH and altered regulons such as MEF2C and JUND. In addition, we reported that one of the endothelial cell clusters with *ACKR1* expression exhibited strong chemotaxis and a weak angiogenic ability and expanded with disease progression. Furthermore, ligand-receptor-based cell-cell interaction analysis indicated that ACKR1+ endothelial cells might specifically communicate with stromal cells through the VISFATIN and SELE pathways, thus influencing stromal cell differentiation in ONFH. Overall, our data revealed single cell transcriptome characteristics in alcohol-induced ONFH, which may contribute to the further investigation of ONFH pathogenesis.

[1] Department of Orthopaedic Surgery, Nanfang Hospital, Southern Medical University, 510515 Guangzhou, Guangdong, China. [2] Guangdong Provincial Key Laboratory of Construction and Detection in Tissue Engineering, Southern Medical University, 510515 Guangzhou, Guangdong, China. [3] General Administration Office, ZhuJiang Hospital of Southern Medical University, 510280 Guangzhou, Guangdong, China. [4] Orthopaedic Department, The 8th medical center of Chinese PLA General Hospital, 100091 Beijing, China. [5] Department of Orthopaedic, The 920th Hospital of Joint Logistics Support Force, 650020 Kunming, Yunnan, China. [6] Department of Orthopaedic Surgery, Shunde First People Hospital, 528300 Foshan, Guangdong, China. [7] These authors contributed equally: Zheting Liao, Yu Jin, Yuhao Chu, Hansen Wu. ✉email: tanhongbo021@yeah.net; lzhaonf@126.com

Nontraumatic osteonecrosis of the femoral head (ONFH) is a disabling orthopaedic disease that is pathologically characterized by femoral head microvascular dysfunction and bone metabolism disorder. Alcoholism is the main pathogenic factor in nontraumatic ONFH and is responsible for 32.4–45.3% of nontraumatic ONFH cases in Asia[1–3]. Although many hypotheses exist regarding the pathogenesis of alcohol-induced ONFH, the exact pathogenesis has remained unclear.

In current clinical practice, non-replacement procedures, such as vascularized bone flap grafts and stem cell-loaded tantalum grafts, can preserve hip function in middle-stage alcohol-induced ONFH (ARCO stage 2 to 3A). However, the effectiveness of these procedures is uncertain[4–6]. Mesenchymal stem cells (MSCs), which are important seed cells for bone formation, have the ability to undergo self-renewal and differentiate into multiple cell types. The abnormal osteogenic-adipogenic differentiation of MSCs and excessive fat accumulation in lesions of the bone marrow cavity are highly relevant to ONFH[7]. Alcohol has been reported to alter the osteogenic-adipogenic differentiation of MSCs, enhance their adipogenic competency by regulating Wnt and mTOR pathways[8–10] and retard osteogenic differentiation[11,12]. However, whether local MSCs exhibit these characteristics in alcohol-induced ONFH and whether the microenvironment is still conducive to bone regeneration remain unknown.

Alcohol consumption causes endothelial injury and leads to intravascular coagulation, resulting in a reduced blood supply, which has been considered a pathological mechanism of ONFH[13,14]. In contrast, another study reported that endothelial activation-related markers, such as vWF and FVIII, but not coagulation markers, were positively correlated with ONFH progression[15], which indicated the complicated role of endothelial cells in ONFH. Further investigation of the phenotypes and transcriptome characteristics of local endothelial cells will contribute to a better understanding of this phenomenon.

Single-cell RNA sequencing (scRNA-seq) is a powerful tool for elucidating cellular status in bone and bone marrow samples[16,17]. Here, we analysed femoral head cells from patients with middle- and advanced-stage alcohol-induced ONFH (Association Research Circulation Osseous (ARCO) 3A and 4) using scRNA-seq. Samples from patients with hip osteoarthritis (HOA) with chronic joint degeneration and femoral neck fracture (FNF) characterized by acute ischaemia and acute inflammatory responses were used as controls. Our data revealed heterogeneous cellular transcriptome characteristics, identified pathogenic endothelial cells, and explored specific cell–cell communication patterns in alcohol-induced ONFH.

## Results

**Identification of cells in alcohol-induced ONFH.** A freshly isolated single-cell suspension from the femoral head was obtained by even sampling (Fig. 1a and Supplementary Fig. 1), mixed enzyme digestion and density gradient centrifugation (Fig. 1a). Prior to further analysis, we performed quality control of the raw data and excluded cells with low data quality (Supplementary Fig. 2). Thereafter, 46,904 cells in the ONFH group were aggregated into 36 distinct clusters through unbiased clustering and were roughly defined as four major subgroups: myeloid cells, lymphocytes, endothelial cells (ECs), and stromal cells (SCs) (Fig. 1b). In most clusters, the cells from each sample were evenly distributed (Fig. 1c). Myeloid cells and lymphocytes were the most abundant cells, accounting for 55.7% and 24.3% of the total cells, respectively (Fig. 1c). Differentially expressed gene (DEG) analysis was carried out on four major subgroups. The expression patterns of the top 20 DEGs identified in individual cells in these major subgroups were centralized and included recognized cell-type markers (Fig. 1d).

DEG analysis was also carried out on the basis of cell clusters, and the top 20 DEGs and the precise annotations of each cluster are listed in Table S1. Using the same screening criteria and clustering and annotation methods, we identified 23,835 HOA cells and 9,859 FNF cells (Supplementary Figs. 3, 4). Notably, a cluster of haematopoietic cells showing high *HBA1*, *HBA2*, *HBB* and *HBD* expression was only found in HOA.

**Characterization of SCs in alcohol-induced ONFH.** Stromal cell dysfunction is one of the pathological features of alcohol-induced ONFH, so we focused on SCs for in-depth analysis. A total of five clusters of SCs, fibrochondrocytes (FCs), uncommitted stromal cells (USCs), adipogenic lineage cells (ALCs), osteogenic lineage cells (OLCs) and chondrogenic lineage cells (CLCs), were identified by unbiased clustering (Fig. 2a), canonical markers from DEGs (Fig. 2b and Supplementary Data 1). Of note, some cluster feature genes were also expressed at lower levels in other lineages. The osteogenic marker *IBSP* was also expressed in CLCs, the adipogenic marker *APOE* in USCs and OLCs, the stemness feature *NOTCH3* in ALCs and FCs, and feature genes of FCs *COL1A1* and *COL3A1* also in ALCs and CLCs, indicative of the multiple functions of these genes and the cell heterogeneity persisting in the cluster.

Since these five types of SCs are known to show correlations in their biological differentiation[18,19], we conducted a pseudotime analysis to further verify the annotations. The results showed that USCs were near the origin of pseudotime and gradually underwent three cell fates: adipogenesis, osteogenesis and chondrogenesis (Fig. 2c). Some of the CLCs and OLCs constituted the prebranch of osteogenesis, and most of the FCs were located at the end of the chondrogenesis branch. Although most of the cells in each cluster had a tendency to concentrate towards a pseudotime node, some of the cells were also scattered in the prebranch. The expression peaks of recognized marker genes appeared in different stages of pseudotime, which was consistent with the cell fates (Fig. 2d). Moreover, DEG analysis of the two trajectory nodes showed that the top 10 DEGs contained recognized lineage-specific genes (*ADIRF, IBSP, SPP1*) with expression trends that were consistent with the cell fates (Fig. 2e). Some genes (*TPM2, SPARCL1, OLFML3, PTGDS, EFEMP1, HAPLN1*) lacked a reported connection with MSC differentiation also showed different expression levels among each cell fate. We also carried out Gene Ontology (GO) enrichment analysis of the highly expressed DEGs of the five SCs and found that corresponding cell fate terms were enriched in each cluster and that each term contained more than 10 DEGs (Fig. 2f).

To further explore the transcriptome characteristics of the five types of SCs, we performed regulon specificity analysis and regulon activity analysis of the SCs (Fig. 2g, h). The results showed that each type of SC exhibited specific and highly active regulons, including recognized MSC fate determinant regulons CLCs: SOX9 (13g) and SOX9_extended (14g); ALCs: CEBPA_extended (34g); OLCs: SP7_extended (61g); USCs: NR2F2_extended (71g)) as well as poorly defined regulons (FCs: BHLHE40_extended (184g) and CREB3L1_extended (162g)) (Fig. 2g, h).

**Comparison of osteogenic- adipogenic differentiation between groups**. We next compared SCs in different stages of ONFH, HOA and FNF to explore the transcriptional characteristics specific to ONFH. By using canonical marker genes, the merged stromal cells could also be annotated as chondrogenic-, adipogenic-, uncommitted- and osteogenic- stromal/lineage cells, and a cluster of undefined cells lacking stromal marker genes (Fig. 3a). Notably, the FCs in the ONFH group were included in CLCs after multigroup integration. In terms of cell proportions, the ONFH 3A group

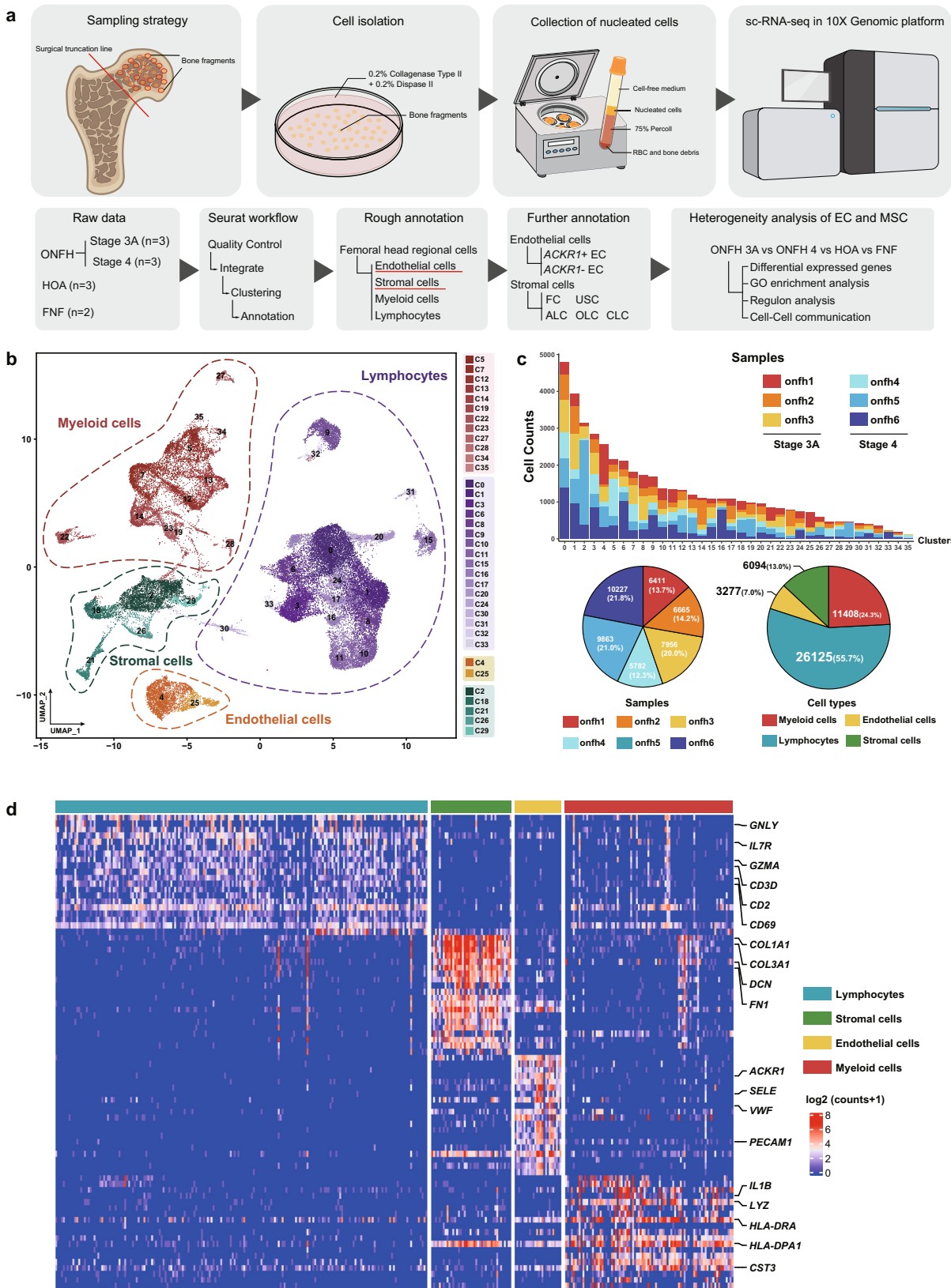

**Fig. 1 Landscape of local cells on alcohol-induced ONFH femoral heads. a** Schematic diagram of single-cell suspension acquisition and sequencing data analysis. **b** UMAP plot of 46,904 high-quality primary cells on alcohol-induced ONFH femoral heads. The cells were grouped into 36 clusters consisting of 4 major groups of cells. **c** Bar plot and pie charts showing the cell composition within each cell cluster and the total ONFH cells. **d** Heatmap showing the top 20 DEGs of the 4 major cell subgroups. The genes labelled on the right are recognized cell-type markers.

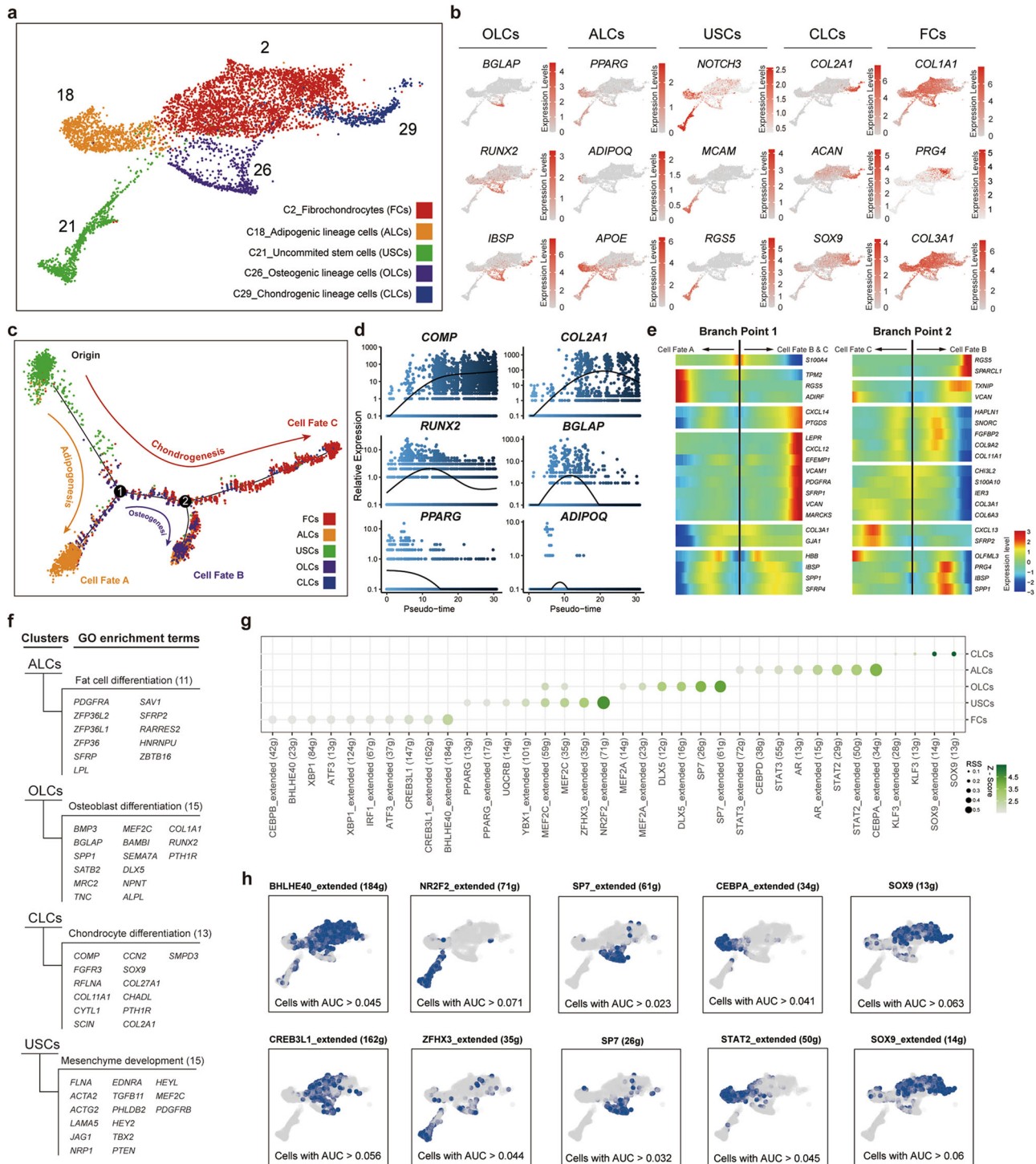

**Fig. 2 Characterization of stromal cells in alcohol-induced ONFH. a** UMAP showing the clustering and annotation of stromal cells. **b** Feature plots showing the distribution of recognized lineage-specific genes. **c** Trajectory map indicating the developmental correlations of the 5 SC clusters. **d** Lineage-specific gene expression changes over pseudotime. **e** Heatmap showing the top 10 DEGs between different cell fates at development node 1 and node 2 in (**c**). **f** Differentiation-related GO terms enriched by the top 100 DEGs of each SC cluster. The numbers and names of the enriched genes are listed behind and below the GO terms, respectively. **g** Bubble plot showing the Regulon specificity score (RSS) of each SC cluster's highly specific regulons. **h** Feature plots showing the binary regulon activity scores of the two regulons with the highest RSSs in each SC cluster. AUC: area under the curve.

contained the highest proportion of ALCs, while the ONFH 4 group showed the highest proportion of CLCs (Fig. 3b). The proportion of USCs in the FNF group was higher than that in the other groups (Fig. 3b). H&E staining showed that the bone marrow in the HOA and FNF groups was full of adipose tissue, in which adipocytes were mature and had large lipid droplets (Fig. 3c). Conversely, the bone

marrow of the ONFH group was filled with fibrous materials and a large number of small vacuolar cells (Fig. 3c). To identify these small vacuolar cells, we selected chemerin, a known adipokine[20] specifically expressed in the ALCs in our sequencing data (Supplementary Fig. 5a), as a marker for immunohistochemistry detection. High levels of chemerin-positive signals were found

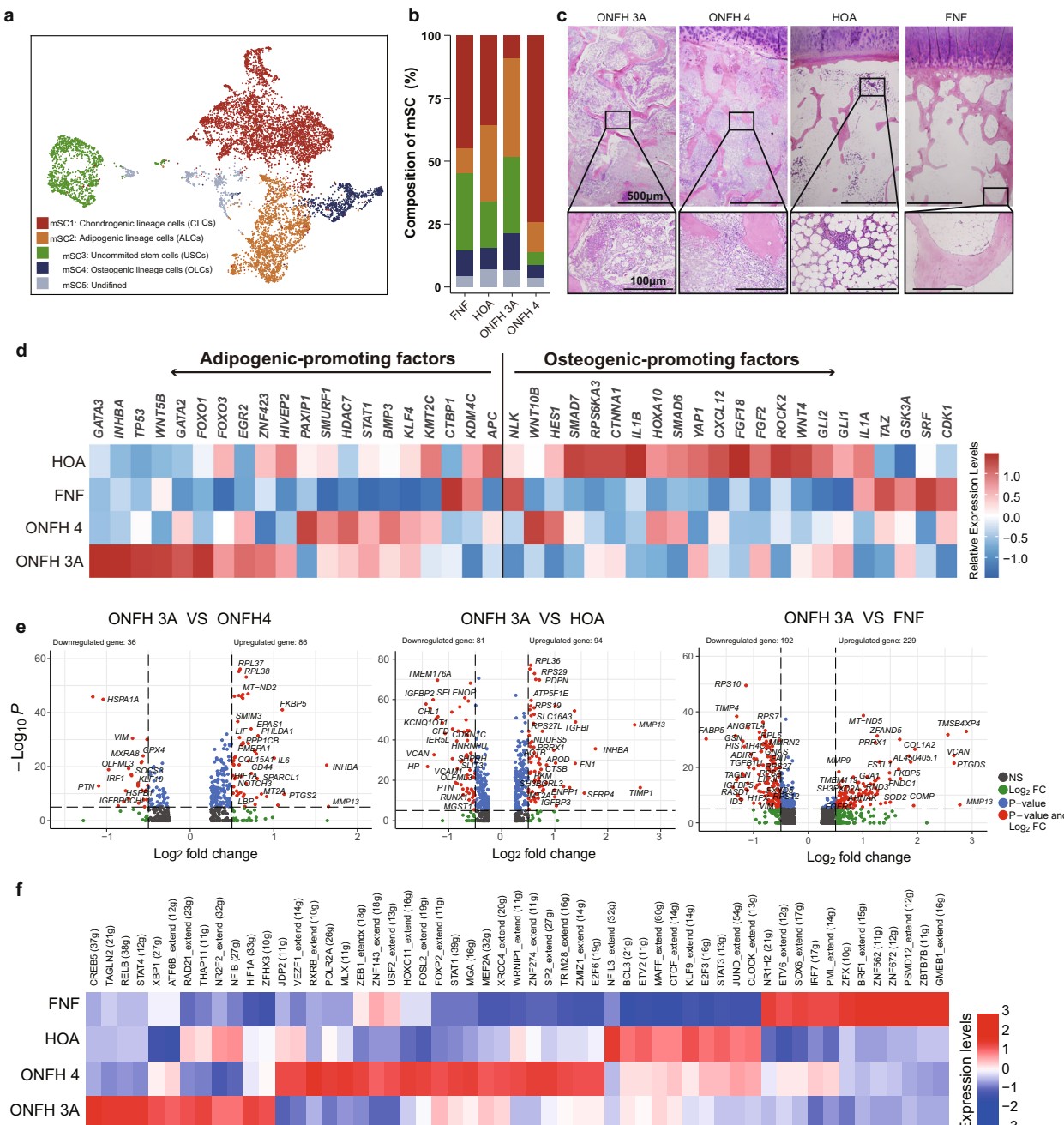

**Fig. 3 Diversity of osteogenic- and adipogenic- trends among the ONFH, HOA, and FNF groups. a** UMAP showing the clustering of stromal cells integrated by the ONFH (stage 3A and 4), HOA and FNF groups. **b** Bar plots showing the constituent ratio of each merged stromal cell (mSC) cluster. **c** Representative haematoxylin and eosin-stained images of femoral head specimens. The lower image is an enlargement of the black box area in the upper image. Scale bar in the upper image = 500 μm, scale bar in the lower image = 100 μm. **d** Heatmap showing the relative expression levels of recognized osteogenic-promoting and adipogenic-promoting factors in the collection composed of USCs, OLCs and ALCs from different groups. **e, f** DEGs ($P < 0.05$, log$_2$ fold change > 0.5) and regulon relative activity of USCs+OLCs+ALCs from different groups.

around these cells (Supplementary Fig. 5b), indicating that these cells were adipocyte lineage cells.

We then investigated the differences in osteogenic-adipogenic differentiation of SCs between different groups from scRNA-seq data. The expression levels of adipogenic promoters[21–24] were higher in the ONFH 3A group, while the expression levels of osteogenic promoters were higher in the HOA and FNF groups in the collection composed of OLCs, ALCs and USCs (Fig. 3d), USCs alone (Supplementary Fig. 6a) or other collections (Supplementary Fig. 6b, c). The GO and Kyoto Encyclopedia of

Genes and Genomes (KEGG) enrichment analyses of the DEGs identified in USCs showed that the adipogenesis term appeared in the ONFH group, the skeletal development term appeared in the HOA group, and the autophagy and apoptosis terms appeared in the FNF group (Supplementary Fig. 6d). To verify the diverse adipogenic-osteogenic competency of the SCs in each group, we isolated and amplified MSCs from ONFH 3A, HOA and FNF specimens (Supplementary Fig. 7a–c). Differentiation experiments confirmed that MSCs in the ONFH 3A group showed a stronger adipogenic differentiation ability (Supplementary Fig. 7d)

and a weaker osteogenic differentiation ability (Supplementary Fig. 7e).

To further explore the underlying molecular mechanisms, we regarded the cells with osteogenic- or adipogenic- differentiation potential (USCs, ALCs and OLCs) as a collection and compared DEGs and regulon activity between groups (Fig. 3e, f). NOTCH3, MMP13, MEF2C (27g), ZNF282 (12g), and SMACB1_extend (11g) were found to show higher expression levels or regulon activity in the ONFH 3A group. The activity of transcription factors known to promote osteogenic differentiation, such as JUND_extend (54g), SOX6_extend (17g) and PML_extend (14g), was lower in the ONFH group (Fig. 3f).

**Characterization of ECs in alcohol-induced ONFH.** Endothelial cell dysfunction is another important pathological feature of alcohol-induced ONFH. Here, we further investigated the transcriptional characteristics of endothelial cells. Two distinct endothelial cell clusters were identified in the ONFH group by unbiased clustering (Fig. 4a). *ACKR1* was the most differentially expressed gene between these two clusters. Therefore, we referred to the two clusters as ACKR1+ ECs and ACKR1− ECs. The DEG analysis showed that the ACKR1+ ECs overexpressed *ACKR1*, the blood coagulation-related gene *PLAT*, and the immune adhesion-related gene *SELE* (Fig. 4b). The ACKR1- ECs highly expressed the angiogenic-related gene *SEMA3G*, the chemokine gene *CXCL12*, and the tight junction component *CLDN5* (Fig. 4b). To further characterize these two clusters of ECs, GO analysis was performed. Over 30 of the top 100 GO terms enriched in ACKR1+ ECs were associated with the response to inflammation and chemokines, leucocyte adhesion and migration, and the response to hypoxia and oxidative stress (Fig. 4c and Supplementary Data 2). The ACKR1− ECs were enriched in GO terms (over 20 of the top 100) related to endothelial cell development, differentiation, cell junctions and vasculature development (Fig. 4c and Supplementary Data 2).

Heterogeneity among these two endothelial cell clusters was also observed in the regulon analysis. NR2F2_extended (171g), JUND_extended (990g), and BHLHE40_extended (6103g) constituted the top cell type-specific regulons in ACKR1+ ECs (Fig. 4d, e). Conversely, ACKR1-ECs possessed the specific regulons SOX17_extended (22g), KLF3_extended (27g), PPAR-G_extended (44g) and ETS1 (84g) (Fig. 4d, e).

To verify the heterogeneity among ACKR1+ ECs and ACKR1− ECs, endothelial cells derived from alcohol-induced ONFH specimens were isolated and amplified. ACKR1+/− ECs were separated by flow cytometry (Fig. 4f and Supplementary Fig. 8). A tube formation experiment was then performed to evaluate the angiogenesis ability of endothelial cells. The results showed that the ACKR1- ECs generated more branch points and exhibited longer total tube lengths than the ACKR1+ECs (Fig. 4g, h). To assess the chemotaxis and leucocyte extravasation ability of the ECs, we used two different Transwell cell migration models (Fig. 4i). In the initial Transwell model, a monolayer of ACKR1+ ECs in the upper chamber resulted in more downwards-migrating THP-1 cells (a human peripheral blood monocyte cell line) than monolayers of ACKR1− ECs and unsorted ECs (Fig. 4j, k). In another model, more THP-1 cells were recruited to the conditioned medium of ACKR1+ ECs than to the conditioned medium of ACKR1− ECs or unsorted ECs (Fig. 4j, k). In summary, these data suggested the presence of two heterogeneous clusters of ECs in alcohol-induced ONFH, among which ACKR1+ ECs presented a proinflammatory phenotype.

**Heterogeneity of ECs in the ONFH, HOA and FNF group**. In contrast to the ONFH group, three clusters of endothelial cells

were found in the HOA group, which could also be distinguished into ACKR1+ ECs and ACKR1− ECs, while the unbiased clustering of the FNF group yielded only one cluster of ACKR1-endothelial cells (Supplementary Figs. 3, 4). After the integration of ECs from ONFH, HOA and FNF, we divided these cells into ACKR1+ ECs and ACKR1− ECs depending on their *ACKR1* expression levels (>3 counts, ACKR1+ ECs; ≤3 counts, ACKR1− ECs) (Fig. 5a). In the merged ECs, the number of ACKR1+ ECs varied by disease and status; ACKR1+ ECs accounted for 3.95% of FNF ECs, 34.88% of ONFH 3A ECs, 48.61% of ONFH 4 ECs (39.85% of total ONFH ECs), and 43.57% of HOA ECs (Fig. 5b). The expression level of *ACKR1* was higher in the ONFH 4 group than in the HOA, ONFH 3A and FNF groups (Fig. 5c). ACKR1+/vWF+ cells were mainly observed in sinusoids and venules, but not in arteries (Fig. 5d). The number of ACKR1+ vessels and ACKR1+ primary endothelial cells in the ONFH 4 group was greater than that in the other groups (Fig. 5d, e), which was consistent with the number of ACKR1+ ECs in the single-cell sequencing data.

We further performed intergroup comparisons of ACKR1+ ECs and ACKR1- ECs, respectively. Among the ACKR1+ ECs, the ONFH and HOA groups showed higher expression levels of *ACKR1*, leucocyte adhesion molecules (*SELE*, *SELP*, and *ICAM1*) and inflammatory factors (*IL1R1*), while an opposite trend of angiogenesis-related gene (*KDR*, *CDH5*, *ITGB1*, *CLDN5*, and *FABP4*) expression was observed in ACKR1- ECs (Fig. 5f). To clarify the relationship between ACKR1 expression and the progression of alcohol-induced ONFH, we established a mouse chronic alcohol consumption model by administering an alcohol-containing liquid diet. The expression levels of ACKR1 in the femoral head of the model mice increased gradually over time, and the ACKR1 signal in the 7th week was significantly higher than that in the control group (Fig. 5g, h). To further explore the transcriptome differences of ACKR1+ EC in different statuses, we conducted regulon analysis. Regulons related to inflammation and hypoxia, such as STAT1 (35g), STAT2 (17g), and ATF4_extended (48g), showed higher activity in the ONFH group, while regulons related to endothelial survival and tight junctions, such as ETS2_extended (18g) and RXRA (24g), showed higher activity in the HOA group (Fig. 5i). Taken together, these data suggested that there was heterogeneity in the transcriptome characteristics of ACKR1+ ECs in different groups and that ACKR1+ ECs might play a pathogenic role in alcohol-induced ONFH.

**Analysis of EC and SC communication reveals potential regulatory pathways in alcohol-induced ONFH.** The above data revealed the specific transcriptome characteristics of SCs and ECs in alcohol-induced ONFH, and we then tried to identify the possible underlying pathogenic mechanism from the perspective of cellular communication between SCs and ECs. In the ligand-receptor pair-based communication analysis of all clusters in ONFH, it was found that ECs harboured high-ranking ligand numbers, which were paired with receptors from myeloid cells and SCs (Fig. 6a). On the other hand, C22 (annotated as osteo-clasts, expressing *CTSK*, *MMP9*, and *ACP5*) harboured the most receptors, which were paired with ligands from ECs and SCs (Fig. 6a). Overall, ligand-receptor pairs were mainly enriched between SCs, ECs and some myeloid cells (Fig. 6a red frame), suggesting a high frequency of cell communication between these cells. High-frequency cell communication between ECs and SCs was also observed in the HOA and FNF groups (Supplementary Fig. 9). We then analysed the main communication patterns and corresponding highly weighted pathways among all clusters (Fig. 6b and Supplementary Fig. 10a–d). Notably, several stem cell

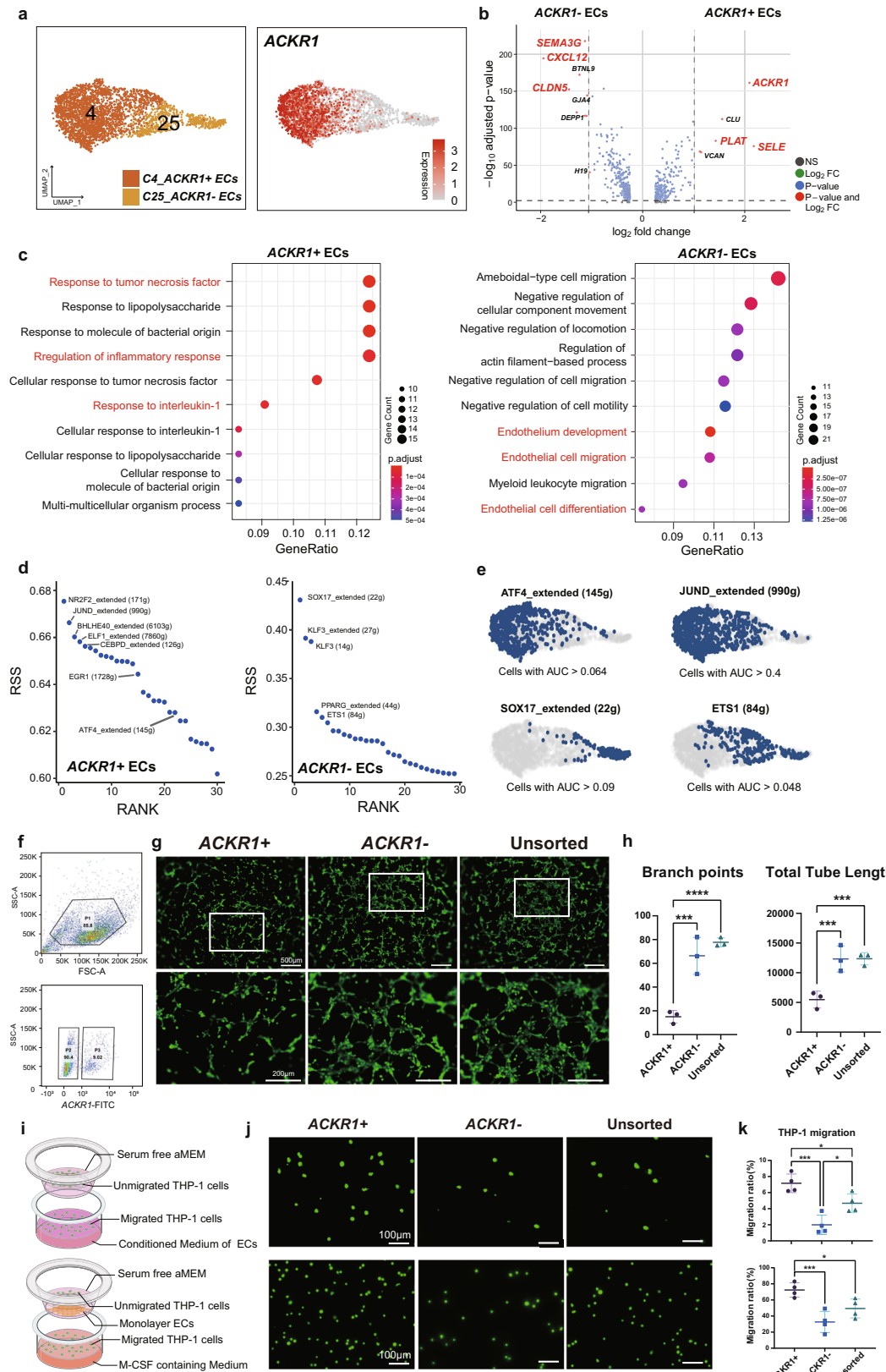

differentiation-related pathways, such as the BMP, FGF and PDGF pathways, were primarily found in ECs, which indicated that communication between ECs and SCs might play a role in SC differentiation in alcohol-induced ONFH.

The above data revealed two distinct clusters of ECs with different transcriptome characteristics in alcohol-induced ONFH,

and we then compared the communication characteristics between these two clusters of ECs and SCs. The VISFATIN and SELE pathways were significantly upregulated in ACKR1+ ECs via the NAMPT-(ITGA5/ITGB1), NAMPT-INSR, SELE-CD44 and SELE-GLG1 axes (Fig. 6c). The PDGF, CXCL12, and SEMA3 signalling pathways were significantly upregulated in

**Fig. 4 Two clusters of differentiated endothelial cells expressing ACKR1 were identified. a** The unbiased clustering of ONFH endothelial cells and the expression distribution of ACKR1. **b** DEGs between ACKR1+ ECs and ACKR1− ECs. Genes with a $\log_2$ fold change greater than or less than 1.0 are shown as green points, genes with a $-\log_{10}P$ greater than 5.0 are shown as blue points, and genes that satisfied both conditions are shown as red points. **c** Bubble plots representing the top 10 GO terms ($P < 0.05$) of ACKR1+ ECs (left) and the top 10 GO terms ($P < 0.05$) of ACKR1- ECs. The size of the point represents the number of genes enriched. The colour of the dot represents the adjusted $P$ value. **d** Top cell type-specific regulons of ACKR1+ ECs and ACKR1- ECs based on the regulon specificity score (RSS). **e** Feature plots indicating selected regulon binary activity scores. **f** Flow cytometry gating strategy for ACKR1+/− ECs. P1: nucleated cells; P2: ACKR1− ECs; P3: ACKR1+ ECs. **g** Representative calcein AM-stained images of tube forming assays. The image in the lower row is an enlargement of the white-boxed area in the upper row. Scale bar in the upper row= 500 μm, scale bar in the lower row= 200 μm. **h** Quantitative analysis of branch points and total tube length. The analysis was based on the angiogenesis analysis plugin in ImageJ software. Data are represented as mean ± SD, $n = 3$. One-way ANOVA with Tukey's test, $***P < 0.001$, $****P < 0.0001$. **i** Schematic diagram of two Transwell experiments. **j** Representative calcein AM-stained images of migrated THP-1 cells. Scale bar = 100 μm. **k** Migration ratio of THP-1 cells in different Transwell experiments based on automatic cell counting. Data are represented as mean ± SD, $n = 4$. One-way ANOVA with Tukey's test, $*P < 0.05$, $***P < 0.001$.

ACKR1- ECs via the SEMA3F-(NRP2 + PLXNA1), SEMA3G-(NRP2 + PLXNA1), PDGFB−PDGFRA, PDGFB−PDGFRB, PDGFD−PDGFRB and CXCL12-ACKR3 axes (Fig. 6c). Next, we compared the ECs-SCs communication patterns between groups (Fig. 6d). Overall, the cell communication patterns of ACKR1+ ECs and ACKR1− ECs to SCs were different in each group. The VISFATIN (secreted) and SELE (cell–cell contact) pathways were the main bridges of ACKR1+ ECs − SCs communication in the ONFH group. In the HOA group, the MK, MIF (secreted), APP and SELE (cell–cell contact) pathways accounted for a high proportion. In the FNF group, there was an elevated ANGPTL (secreted) pathway.

To further forecast the genetic functions of these differentially expressed pathways, Pearson's correlation analysis was performed between these receptors and classical osteogenic, adipogenic and chondrogenic marker genes in stromal cells (Fig. 6e and Supplementary Fig. 10e). In the ONFH group, receptor genes *ITGA5*, *ITGB1* and *CD44* were negatively correlated with osteogenic and adipogenic marker genes but positively correlated with chondrogenic marker genes. The receptor genes *INSR* and *GLG1* showed no tendency during the statistical analysis. In the HOA group, we also found a similar tendency but with a weaker correlation coefficient in SCs (Supplementary Fig. 10e). The above data suggested that the SELE and VISFATIN pathways had an effect on SC differentiation.

## Discussion

Fatty hyperplasia, abnormal bone metabolism and microvascular dysfunction are considered to be common pathological changes in alcohol-induced ONFH. MSCs are also the cellular basis of stem cell therapy in the clinical treatment of alcohol-induced ONFH[5,9]. Thus, endothelial cells, mesenchymal stem cells and osteoblasts have become the focus of research on the mechanism of alcohol-induced ONFH development[25,26]. However, previous studies have used cells that have either been amplified in vitro or isolated from peripheral blood, which cannot fully reflect the transcriptome characteristics of local cells in the femoral head. In our study, we use scRNA-seq to analyse freshly isolated primary cells from alcohol-induced ONFH, which helped to more comprehensively understand the transcriptome characteristics of local cells.

In the SCs of alcohol-induced ONFH, we defined five types of stromal cells in different commitment differentiation trajectories. Notably, a large number of FCs were present in the ONFH group, and this number increased with progression (stage 4 > 3A). The hypoxia-related transcription factors BHLHE40[27] and CREB3L1[28] and their regulons were specifically active in fibrochondrocytes. These findings are consistent with the consensus observation of enhanced SC chondrogenesis and chondrocyte redifferentiation in the hypoxic environment[29,30], which reflects

enhanced hypoxia in the bone marrow microenvironment during disease progression. Elucidation of the role of proliferative fibrochondrocytes in alcohol-induced ONFH will require further experimental research.

Previous studies have reported that the activity and osteogenic differentiation ability of femoral head-derived MSCs in ONFH are significantly reduced, and our data are consistent with those findings. Pseudotime analysis suggested the presence of early- and middle- differentiated cells in ALCs and OLCs, thus, it is difficult to equate USCs with MSC in biology. Therefore, the USCs+ALCs +OLCs, USCs+ALCs and USCs+OLCs combinations were included in the intergroup comparison when analyzing differentiation trends. In either case, consistently reduced osteogenesis and elevated adipogenesis in the ONFH group was observed. Thus, we further explored the possible molecular mechanisms and found the upregulated osteoclastogenesis-related regulon MEF2C (27g)[31] and downregulated osteogenesis-related regulon JUND_extend (54g)[32], as well as many unverified DEGs and regulons. These data may provide clues for future studies on osteogenic-adipogenic differentiation imbalance in MSCs.

ACKR1/Duffy antigen receptor for chemokines (DARC) is a nonspecific inflammatory chemokine receptor that is widely expressed in erythrocytes and endothelial cells[33,34]. Previous studies have found that DARC in endothelial cells increases leucocyte extravasation and inhibits neovascularization[33,35]. Interestingly, we found that two distinct clusters of endothelial cells in alcoholic ONFH could be clearly distinguished by their expression levels of *ACKR1*. DEG analysis showed that ACKR1+ ECs expressed high levels of leucocyte adhesion- and migration-related genes and lower levels of cellular junction-related genes, which might also contribute to leucocyte chemotaxis and extravasation, in addition to the function of DARC itself. In addition, transcriptional regulon analysis showed that ATF4_extended (145g), which contained the target chemokine gene *CCL2*[36], exhibited specifically high activity in ACKR1+ ECs. ETS1 (84g) and SOX17_extended (22g), which are closely related to angiogenesis[37,38], exhibited lower activity in ACKR1− ECs. Combined with the results of the GO analysis, Transwell assays and tube formation assays, our data suggested that ACKR1+ ECs had a stronger ability to recruit immune cells and a weaker ability to undergo angiogenesis. Thus, ACKR1+ ECs showed an activated phenotype[39,40] and highly expressed proinflammatory cytokines, chemokines and adhesion molecules.

The interdisease comparison showed that ACKR1+ ECs expanded with ONFH progression (stage 4 > 3A). The expression levels of ACKR1 in the femoral head also increased gradually over time in a mouse alcohol consumption model. These results suggested that ACKR1+ EC amplification was associated with ONFH disease progression, but it was not clear whether there was a direct causal relationship between the two, which will be

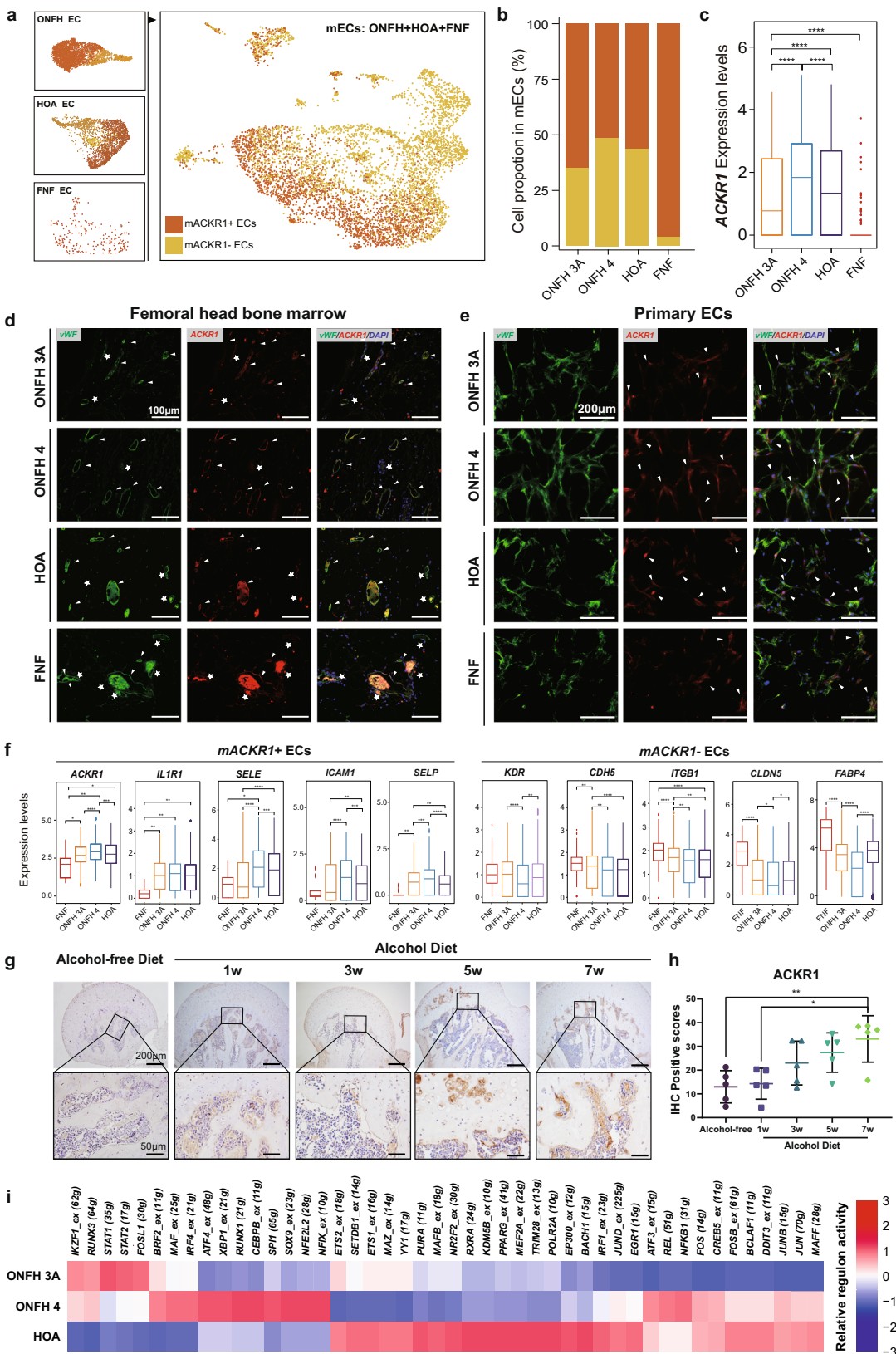

revealed by further studies, such as cell-tracing experiments. Another concern was also a certain amount of ACKR1+ ECs in the HOA group, although the amount and proportion were smaller than those in the ONFH 3A group. The role of ACKR1+ ECs in aseptic inflammation and bone remodelling of subchondral bone in osteoarthritis merits further investigation.

scRNA-seq can generate transcriptome data from multiple cells in the same microenvironment, which provides reliable data for L-R pair-based cell–cell communication analysis[41]. Our data showed that L-R pairs were highly enriched in communication between ECs and SCs in alcohol-induced ONFH. Furthermore, ACKR1+ ECs and ACKR1− ECs showed different pathway

**Fig. 5 Characterization of ACKR1+/− ECs in ONFH, HOA and FNF. a** The distribution of *ACKR1* in the merged endothelial cells in ONFH, HOA and FNF. **b** The proportion of mACKR1+ ECs and mACKR1− ECs in FNF, ONFH 3A, ONFH 4 and HOA. **c** The expression levels of *ACKR1* in each group. Boxplot shows the median and quartile values. T-test, ****$p < 0.0001$. **d, e** Presentative images of the immunofluorescence staining of ACKR1 and vWF in the bone marrow region of femoral head samples (scale bar = 100 μm) and in the cultured endothelial cells (scale bar = 200 μm). Green fluorescence indicates vWF, red indicates ACKR1, and blue indicates DAPI. Arrows indicate ACKR1+/vWF+ vessels/cells, stars indicate ACKR1−/vWF+ vessels. **f** The expression levels of functional genes in mACKR1+ ECs and mACKR1− ECs. Boxplot shows the median and quartile values. T-test, *$P < 0.05$, **$P < 0.01$, ***$P < 0.001$, ****$P < 0.0001$. **g, h** Representative images of immunohistochemical staining of ACKR1 in the femoral head of mice with chronic alcohol consumption and quantification analysis. Scale bar = 200 μm. Data are represented as mean ± SD, $n = 5$, One-way ANOVA with Tukey's test, *$p < 0.05$, **$p < 0.01$. **i** Heatmap showing the relative regulon activity of mACKR1+ ECs in each group. The FNF group was eliminated because of the low number of ACKR1+ ECs.

patterns in communication with SCs. The VISFATIN and SELE pathways showed a high probability of being involved in communication between ACKR1+ ECs and SCs.

Visfatin/Nampt is a ubiquitous essential enzyme with catabolic and proinflammatory properties that have been reported to induce an inflammatory phenotype of fibroblasts in rheumatoid arthritis[42] and to promote chondrocyte apoptosis and extracellular matrix degradation in osteoarthritis[43,44]. Our data suggested that ACKR1+ ECs might communicate with CLCs, OLCs and FCs via the VISFATIN pathway through the NAMPT-INSR/NAMPT-ITGA5-ITGB1 axis. In addition, we found that the communication weights of the VISFATIN pathway were upregulated in the ONFH group compared with the other 2 groups. Correlation analysis further revealed that ITGA5 and ITGB1, the main receptors of the VISFATIN pathway, were negatively correlated with osteogenic marker genes and positively correlated with chondrogenic marker genes. These findings provide promising targets for the study of abnormal MSC differentiation in ONFH.

SELE (Selectin E), another ACKR1+ ECs−SCs communication pathway associated with stem cell differentiation and elevated in the ONFH group, has been reported to serve as a component of the vascular niche that regulates hematopoietic stem cell dormancy and proliferation[45–47]. Since the SELE pathway is a cell–cell contact type, a prerequisite for this pathway to truly work is vascular injury, so ACKR1+ ECs have the opportunity to contact MSCs in the niche. Immunofluorescence staining showed that ACKR1+ ECs were widely distributed in sinusoids and venules in the ONFH group, making it possible for ACKR1+ ECs to contact MSCs during vascular injury caused by alcohol abuse and other pathogenic factors.

Overall, the present study provided high-resolution single-cell transcriptome data from local cells in alcohol-induced ONFH, revealed stromal cell abnormalities and endothelial cell heterogeneity, identified ACKR1+ ECs as key regulatory cells, and revealed relevant cell–cell communication patterns and highly weighted pathways. These data will contribute to the development of cell-targeted therapy for alcohol-induced ONFH.

## Methods

**Subject details**. In this study, a total of 11 femoral head samples from alcohol-induced ONFH (ARCO 3A $n = 3$; ARCO 4 $n = 3$), HOA ($n = 3$), and FNF ($n = 2$) patients were used for the scRNA-seq experiments. The details of the sample donors are listed in Supplementary Table 1. ONFH was diagnosed according to the accepted standard, including a history of drinking at least 300 g of ethanol per week for more than 6 months. All donors signed informed consent forms and were free from systemic immune diseases or infection. This study was approved by the Nanfang Hospital Ethical Medical Committee (ref. NFEC-2020-154).

**Preparation of samples for single-cell RNA sequencing**. The femoral head specimens were cut along the coronal plane with a wire saw. The bone tissue was then collected from the entire coronal plane with a crescent bone-chisel in a 0.5 cm gap (Fig. 1a and Supplementary Fig. 1), washed once with DPBS, immersed in a mixture of 0.2% (w/v) collagenase type II (C6885, Sigma, St. Louis, USA)/0.2% (w/v) dispase (D4693, Sigma), and placed on 37 °C horizontal rotators at 150 rpm for

4 h. After digestion, the bone residue was removed with a 70 μm nylon filter, and the liquid portion was centrifuged at $400 \times g$ for 5 min. The precipitate containing bone and bone marrow-derived cells and numerous small bone fragments was then resuspended in 20 ml of aMEM and placed in a 50 ml centrifuge tube containing 20 ml of a 75% (v/v) Percoll solution (P1644, Sigma). The whole mixture was centrifuged at $450 \times g$ for 20 min. After centrifugation, approximately 5 ml of the nucleated cell layer was collected, diluted with aMEM (1:10), and recentrifuged at $400 \times g$ for 5 min. Precipitated cells were resuspended in DPBS, and automated cell counting (Countstar BioTech, Countstar, Shanghai, China) was performed under a microscope. Samples with a cell viability rate greater than 90% were eventually used for single-cell RNA sequencing.

**Single-cell RNA sequencing**. Approximately 10,000 cells per sample were loaded into microfluidic chips to generate single-cell gel beads in emulsion by using a commercial kit (Chromium Next GEM Single Cell 3′ GEM Kit v3.1, 10X Genomic, Pleasanton, USA,). Subsequent cDNA amplification, quality control and library construction were carried out in accordance with their guidelines. Cell Ranger was used to compare the original FASTQ sequencing data to the reference genome (GRCh38), perform gene expression quantification, and generate a cell-gene expression matrix.

**Single-cell sequencing data analysis**. Quality control and data screening: All individual sequencing data generated by Cell Ranger (the details of the software that we used are listed in Supplementary Table 2) were subjected to quality control by using the R package Seurat. SoupX software was used to detect and remove ambient RNA contamination. Finally, three samples from the HOA group were subjected to the SoupX procedure due to haemoglobin gene contamination and plasma cell contamination. Furthermore, cells with a unique molecular identifier (UMI) number of less than 500, cells expressing fewer than 250 genes, cells with a mitochondrial gene expression ratio greater than 25% and cells with a gene/UMI ratio less than 0.8 were filtered out (Supplementary Fig. 2).

Cell clustering and differentially expressed gene (DEG) analysis: Before clustering analysis, the independent samples from each group were merged, debatched and normalized by using the functions SCTransform and IntegrateData in Seurat. Principal component analysis (PCA)-referenced unsupervised clustering was performed to cluster ONFH cells (PC = 40, resolution = 1.02), HOA cells (PC = 40, resolution = 1.0) and FNF cells (PC = 40, resolution = 0.6). DEG analysis was performed with the FindMarkers and FindAllMarkers functions of Seurat. The former was used to compare the DEGs of two groups of cells, while the latter was used to search for marker genes of all cell clusters and to retain only positive results. The results from the analyses performed with the above two functions with $\log_2$ fold change (FC) values of less than 0.25 were filtered out.

Correlation test: Correlation analyses of the expression levels of two genes in the same cell cluster or the same sample were performed by using the stats package, and Pearson's analysis was selected as the method.

Cell-type identification: In our study, two methods were combined to annotate cell types. The automatic cell annotation software programs SingleR and scCATCH were used for the preliminary annotation of the cell clusters. According to the preliminary annotation results, the cells were divided into four categories: myeloid cells, lymphocytes, SCs and ECs. Thereafter, the cells were further annotated based on published single-cell sequencing data and known classical cell markers.

Pseudotime analysis: Pseudotime analysis was performed using Monocle2 with DDR-Tree and the default parameters. Before single-cell trajectory assessment, Seurat clusters were selected for analysis based on potential intercellular relationships reported in previously published literature, and branch expression analysis modelling (BEAM) was applied for branch fate-determined gene analysis.

GO and KEGG enrichment: Gene enrichment analysis was performed by using the R package ClusterProfiler (Figs. 2f, 4c) and Metascape solfware (Supplementary Fig. 6d). The enrichment analysis of GO and KEGG dual databases was based on the DEGs ($P < 0.05$, $\log_2$FC > 0.25) of the MSCs in each group and was conducted using Metascape. The enrichment of biological process terms was based on the genes that were upregulated in the ACKR1+ ECs and ACKR1− ECs ($P < 0.05$, $\log_2$FC > 0.5) (Fig. 4c).

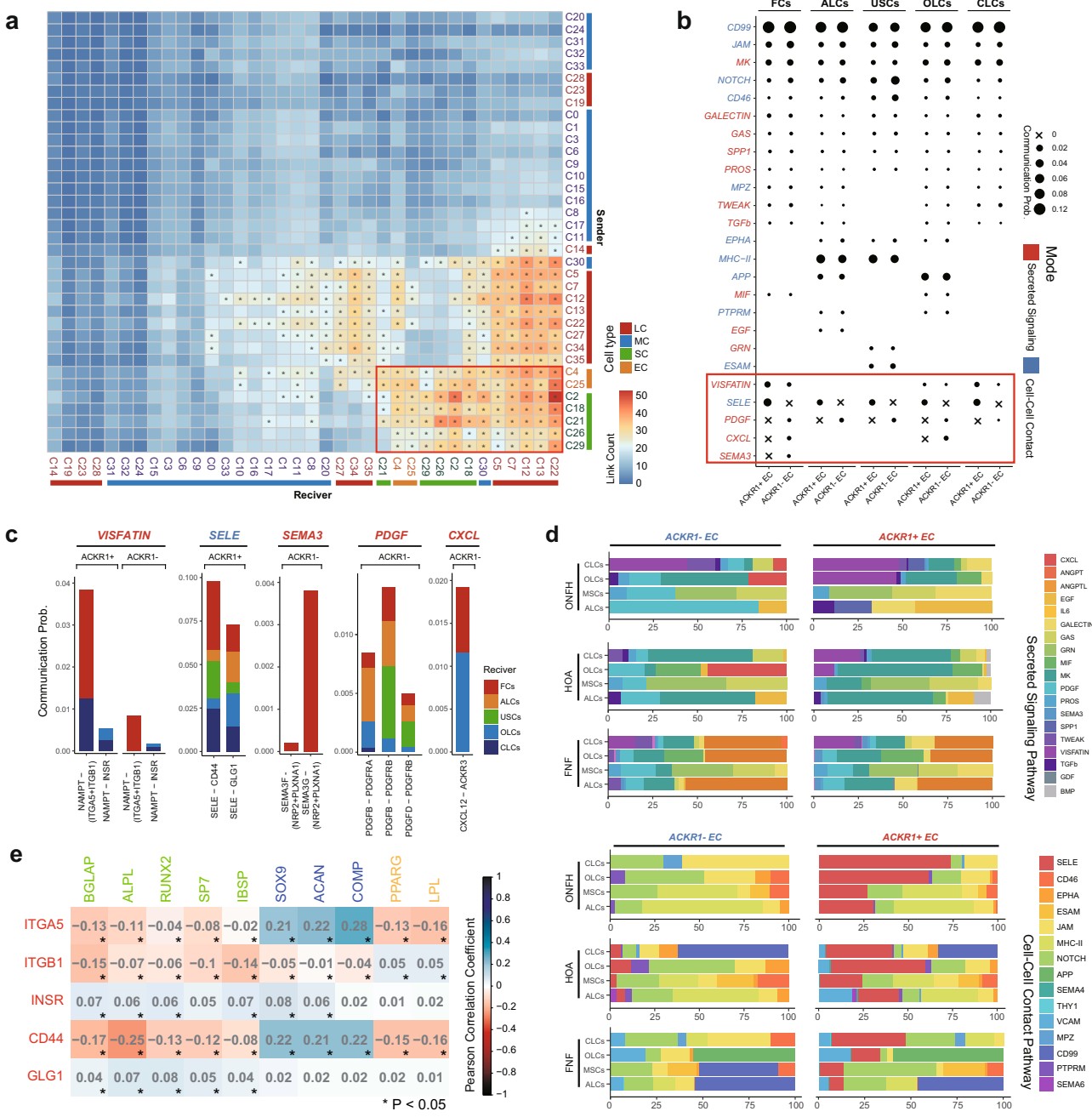

**Fig. 6 Analysis of EC and SC communication indicated potential regulatory signalling pathways in alcohol-induced ONFH. a** Heatmap showing the total counts of significant ligand-receptor pairs across all clusters in the ONFH group. The red frame indicates that EC, SC and myeloid cells harboured high-rank numbers of ligand-receptor pairs. **b** Bubble plot showing specific interaction pathways among ECs and SCs. Pathways in the red frame were markedly different between ACKR1+ ECs and ACKR1- ECs. The bubble size represents the communication probability calculated by CellChat software. Pathway colour represents communication mode (secreted signal or cell–cell contact). **c** Bar plot showing the communication weights and distribution of ligands and receptors from pathways of interest. The colours in the bar column represent receiving cell groups. **d** Bar plot showing the proportions of communication pathways in different groups. The upper panel represents the secreted signalling pathway, and the lower panel represents the cell–cell contact pathway. The colours in the bar column represent communication pathways. **e** Pearson correlation analysis between pathway receptors of interest and adipogenesis or osteogenesis genes. Pearson-test, *P < 0.05.

Analysis of transcriptional regulons: Regulon analysis was performed by using the R package SCENIC and 3 basic packages, GENIE3, RcisTarget and AUCell. First, we used GENIE3 to predict the gene coexpression network of the designated cell clusters. Then, we used RcisTarget to analyse the possible binding sites (motifs) of transcription factors in the target genes. Finally, we used AUCell to score the activity of regulons in each group. After the above preparation, we used SCENIC to further extract and analyse the results and evaluated the RSSs and activities of the regulons.

Cell-to-cell communication analysis: All cell interaction analyses conducted in this study were performed using the R package CellChat. Both secretory communication databases and cell-contact communication databases were used.

**Cell culture and isolation.** The cell suspension obtained via the above method was also used for endothelial cell culture and MSC culture. To culture endothelial cells, regional cells were seeded in 100 mm cell culture dishes at a density of 50,000/cm²

and placed in a 37 °C, 5% CO$_2$ incubator for 1 h. Commercial endothelial cell medium (SC-1001, Cyagen, Santa Clara, USA) was used to culture endothelial cells. After 1 h, unattached cells in the medium were collected, recentrifuged at 350 × g for 5 min, and then reseeded at a density of 20,000/cm$^2$ in 6-well cell culture plates for further culture. When the endothelial cells reached 80% confluence, 0.25% trypsin was used for cell passaging.

Endothelial cells at passage 2 were digested and collected for flow cytometry sorting. The cells were resuspended in DPBS containing 10% goat serum and placed on ice for 20 min. After centrifugation at 350 × g for 5 min, an anti-ACKR1 antibody solution was added, and incubation on ice was conducted for 20 min. Thereafter, the primary antibody was removed by centrifugation at 350 × g for 5 min. The cells were washed twice with DPBS and then resuspended in a FITC-labelled secondary antibody solution for another 20 min. After washing twice with DPBS, the cells were subjected to flow cytometry sorting (Fig. 4f and Supplementary Fig. 8). ACKR1+/−/unsorted ECs were cultured in serum-free aMEM for 3 days, and the remaining aMEM was collected as conditioned medium for ACKR1+/−/unsorted ECs. Information on all antibodies used in this study is provided in Supplementary Table 3.

For MSC primary culture, cells were seeded in 6-well cell culture plates at a density of 200,000/cm$^2$ and placed in a 37 °C, 5% CO$_2$ incubator. aMEM containing 15% foetal bovine serum (FBS) and 1% penicillin and streptomycin (P/S) was used for expansion. MSC purification was achieved by continuous passage in plastic culture plates[48,49]. In short, when the primary adherent cells grew to 70–80% confluence, they were digested with 0.25% trypsin, resuspended in a new medium and seeded into a new culture plate at a density of 2000/cm$^2$. After 2–3 generations of continuous passage according to the above method, the cell morphology was unified into a spindle shape. Cells at passages 3–4 were used for subsequent identification and experiments. To identify MSCs, we examined the expression levels of the classic MSC markers CD105, CD90, and CD73 and the negative markers CD45, CD31, and CD14 by flow cytometry and carried out chondrogenic induction, osteogenic induction, and adipogenic induction experiments as well as cell clone formation experiments (Supplementary Figs. 7, 8).

THP-1 cells (CL-0233), L929 cells (CL-0137) and THP-1 culture medium (CM-0233) were purchased from Procell (Wuhan, China). The THP-1 cells were cultured at 37 °C under 5% CO$_2$, an equal volume of fresh medium was added every 2–3 days, the cells were centrifuged once every 7–10 days, and the cell density was maintained at 50,000 to 200,000/ml. The procedures for the expansion and passaging of the L929 cells were consistent with the MSC method. L929 cells were cultured in serum-free aMEM for 3 days, and the remaining aMEM was collected as L929-conditioned medium.

**Transwell experiments**. Two types of cell migration experiments using the same 8 μm Transwell chambers (3422, Corning, Corning, USA) were carried out in this study. In the first experiment, 20,000 ACKR1+/−/unsorted ECs were seeded on the upper chamber membrane, and endothelial cell culture medium was added to the upper and lower compartments. After 3 days of culture, a monolayer of endothelial cells formed on the membrane. On the 4th day, the upper and lower chambers were washed twice with DPBS, 500 μl of L929-conditioned medium was added to the lower chamber, and 200 μl of a 500,000/ml THP-1 cell suspension was added to the upper chamber. In the second experiment, 500 μl of conditioned medium from ACKR1+/−/unsorted ECs was added to the lower chamber, and 200 μl of a 500,000/mL THP-1 cell suspension was added to the upper chamber.

In both experiments, suspended and adherent THP-1 cells in the lower compartment were collected after 12 h and counted (Countstar BioTech, Countstar) as migrated cells, stained with calcein AM (C2012, Beyotime, Shanghai, China) and observed under a microscope.

**Tube formation assay**. A precooled 96-well cell culture plate was coated with matrix gel (356234, Corning) applied at 50 μl per well and placed in an incubator at 37 °C for 1 h. Thereafter, 50 μl of a suspension containing 15,000 ACKR1+/−/unsorted ECs was added to each gel-coated well. After 6 h of incubation at 37 °C under 5% CO$_2$, the remaining medium was removed, and a calcein AM solution was added for staining for 20 min. The wells were then washed twice with PBS and observed under a fluorescence microscope. The branch points and total tube length in each image were automatically counted using the Angiogenesis Analyzer plugin in ImageJ.

**Multiple differentiation assays**. MSCs at passages 3–4 were seeded in a 12-well cell culture plate at a density of 2000/cm$^2$, and the medium was replaced with osteogenic induction medium (OIM) when the cells reached 70% confluence. The OIM was dexamethasone free and consisted of 50 μg/ml sodium ascorbate (134-03-2, Sigma), 10 mmol/L sodium β-glycerophosphate (13408-09-8, Sigma), 1% P/S and 10% FBS in high-glucose DMEM. The OIM was changed every 3 days, and alizarin red staining was performed on day 14.

In the adipogenesis induction experiment, MSCs were overgrown in 12-well plates, and adipogenesis induction medium (AIM) was then added. Commercial AIM (HUXMA-90031, Cyagen) was used in this experiment. The AIM was changed every 3 days, and oil red/haematoxylin staining was performed on day 14.

In the chondrogenic induction experiment, MSCs were added to a 24-well plate at a density of 200,000/10 μl. After 2 h of culture at 37 °C, the cell drops

became stable. At that time, 1 ml of chondrogenic induction medium (CIM) was added, and the CIM was changed every 3 days. Safranin O staining was performed on day 14. The CIM was high-glucose DMEM containing 10 ng/ml recombinant human TGFβ1 protein (CA59, Novoprotein, Shanghai, China), 100 μg/ml sodium ascorbate, 10 mg/L insulin (I8830, Solarbio, Beijing, China), 1% P/S and 10% FBS. In all differentiation experiments, the control group cells were cultured in high-glucose DMEM containing 10% FBS and 1% P/S, and the medium was changed every 3 days.

**Clone formation experiment**. MSCs at passage 3 were seeded in a 6-well cell culture plate at 100 cells or 50 cells per well and cultured in aMEM containing 15% FBS and 1% P/S. The medium was changed every 3 days. On the 14th day, 1% crystal violet staining was performed to observe the formation of cell clones.

**Animal experiment**. A total of 30 male 8-week-old BALB/c mice weighing between 26 g and 32 g were used to explore the relationship between bone marrow ACKR1 expression and alcohol intervention. This study was approved by the Nanfang Hospital Animal Ethics Committee (ref. NFYY-2020-47). All mice were fed adaptively for one week before the intervention, and the solid forage was gradually replaced with an alcohol-free Lieber- DeCarli liquid forage. All mice were then randomly divided into the alcohol-free group ($n = 6$) and alcohol group ($n = 24$). The alcohol-free group was given nonalcoholic liquid feed and was sacrificed after 7 weeks. The alcohol group was given an alcohol-containing Lieber-DeCarli liquid feed (TP4030, Dyets, Bethlehem, USA), and mice were randomly sacrificed at weeks 1, 3, 5 and 7. Femurs from all mice were isolated for further examination. During the whole experiment, the feeding environment was maintained at a temperature of 20–25 °C and relative humidity of 40–60%, with a normal day-night cycle. All mice were weighed once per week, and mice that had lost more than 15% of their body weight since the last measurement were excluded.

**Histochemistry assays**. Prior to the assay, all femoral head samples were fixed in 4% paraformaldehyde solution at room temperature for 72 h and subsequently subjected to decalcification in 0.5 M EDTA for 16 weeks prior to dehydration, paraffin embedding and serial sectioning (mouse femoral samples were fixed in 4% paraformaldehyde for 12 h at 4 °C and decalcified for 4 weeks). The samples were cut into paraffin sections (4 μm) for subsequent hematoxylin and eosin staining with a commercial staining kit (DM0064, Boster Biological Technology, Wuhan, China) according to the manufacturer's instructions. For immunofluorescence (IF) staining, prepared sections were incubated with anti-ACKR1 and anti-vWF antibodies. FITC- or Cy3-conjugated secondary antibodies were used to display the signals. For immunohistochemistry (IHC), a commercial HRP-DAB system kit (SV0002, Boster) was used to display the signal of the primary antibodies. Information on all antibodies is shown in Supplementary Table 3.

**Statistics and reproducibility**. In cell experiments, each group contained more than 4 biological repeats, and the results for each sample were from the average of 5 different microscope fields. Analysis of the signal strength in the IHC images was performed using the IHC Profile plugin in ImageJ, and the positive score of each image was recorded and used for statistical analysis. In the quantitative analysis, each group contained more than 5 independent samples, and the results for each sample came from the average of three different microscope fields. Independent T-test were used for differential analysis. One-way ANOVA with Tukey's test were used to make pairwise comparisons among multiple groups.

**Reporting summary**. Further information on research design is available in the Nature Research Reporting Summary linked to this article.

## Data availability

All data, including reagents and software information associated with this study are in the paper or the Supplementary Materials. The raw data of scRNA-seq were posted to the SRA database (SRP361778). The processed sequencing data and the source data for graphs are available on Figshare (https://doi.org/10.6084/m9.figshare.19243722).

## Code availability

All analysis scripts are publicly available via: https://github.com/ZhaoLab-JinYu/scRNA-seq_ONFH

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

## Acknowledgements

The authors would like to thank Dr. Li Qiang from Guangzhou Yuanxin Biotechnology for his advice on single-cell RNA sequencing. This study was supported by the following grants: National Key Research and Development Program of China 2017YFC1103900, 2018HB001 (H.B.T.), National Natural Science Foundation of China 31771051 (L.Z.), Natural Science Foundation of Guangdong Province 2018B030311041 (L.Z.), Science and Technology Program of Guangzhou 201803010114 (L.Z.).

## Author contributions

Conceptualization: H.B.T., L.Z. Methodology: Z.T.L., Y.J., H.S.W., Y.H.C., L.Z. Investigation: Z.T.L., Y.J., Y.H.C., H.S.W., X.Y.L., Z.H.D., S.H.F., N.C.C., Z.H.L., X.Y.Z., L.X.B. Visualization: Z.T.L., Y.J., Y.H.C. Supervision: H.B.T., L.Z. Writing—original draft: Z.T.L., Y.J., Y.H.C. Writing—review & editing: L.Z.

## Competing interests

The authors declare no competing interests.
