## [Peer Review File · Communications Biology]

Reviewers' comments:

Reviewer #1 (Remarks to the Author):

This is a very excellent study with the state-of-the-art scRNA-seq technology and fruitful statistical analyses of human stromal and endothelial cells in alcohol-induced ONFH. Liao et al. identified the heterogeneity of stromal and endothelial cells and revealed the differences of their transcriptional information between the case and control groups. In addition, they further illustrated the potential regulatory pathways related to alcohol-induced ONFH through the cell-cell interaction analyses. I have only one suggestion which might help to improve the presentation of the work:

In line 233, the author demonstrated that "communication between ECs and SCs may play a role in SC differentiation in alcohol-induced ONFH" only based on the cell-cell interaction analyses in alcohol-induced ONFH group. It would be more convincing with the comparison of the cell-cell communications of ECs and SCs between case and control groups.

Reviewer #2 (Remarks to the Author):

This paper by Liao et al. reports single cell RNA-seq results obtained from bone specimens of patients with osteonecrosis of the femoral head (ONFH). The authors describe the differences between the single cell bone transcriptomes of patients with ONFH and hip OA or femoral head fracture (as control). One of their primary findings is an enrichment for ACKR1+ endothelial cells in necrotic samples, which the authors suggest could be associated with pro-inflammatory events in the bone marrow micro-environment. Overall, the experiments appear to be carefully conducted and the data rigorously analyzed. I also found the manuscript to be very well-written. I have only a few comments and questions that I am hoping the authors will address:

The data presented in Figure 3c and 3d is seemingly contradictory: While ONFH samples are enriched for transcripts associated with adipogenesis, HOA and FNF specimens have more adipocytes. How do the authors interpret this?

The methodology of MSC purification and culture is a bit unclear: Were these cells sorted with respect to the aforementioned markers (CD105+CD90+CD73+CD45-CD31-CD14-) and then cultured? Have you checked whether these markers are sufficiently specific to the MSC clusters you identified with scRNA-seq?

The stained bone marrow images are difficult to interpret at the current resolution. Could you perhaps provide higher res images and/or individual channels next to each other so that the overlap (or lack of) between vWF and ACKR1 is more obvious? Further, the appearance of ONFH specimens seem to suggest that arteriole-type vessels were labeled – is this correct? Some comments on this in the main body would be helpful to the reader.

Reviewer #3 (Remarks to the Author):

The authors isolated tissue samples from the femoral head of mid- and advanced-stage alcohol-induced osteonecrosis of the femoral head (ONFH) and age-matched hip osteoarthritis (HOA) and femoral neck fracture (FNF) femoral heads. Cells were isolated from the tissue samples and single-cell RNA sequencing (scRNA-seq) was used to compare their transcriptomes. Based on expression profiles and standard bioinformatics approaches, the authors first conclude that there are different stromal cell (SC) types [fibrochondrocytes (FCs), adipogenic lineage cells (ALCs), MSCs, osteogenic lineage cells (OLCs) and chondrogenic lineage cells (CLCs)] that they positioned into a differentiation sequence with MSCs at origin and branches leading to adipogenesis, osteogenesis and chondrogenesis. Comparing ONFH, HOA and FNF samples led to the conclusion that the proportion of the different SC types is different in the 3 conditions and that SC adipogenic and osteogenic differentiation capacity reflected these proportions. None of this is new or unexpected. What is more interesting are the analyses of ECs in the samples, with different proportions of

ACKR1+ ECs and ACKR1- ECs in the three conditions and further analysis suggested that the ACKR1+ ECs, with a more pathological or proinflammatory phenotype, might affect MSC differentiation via the SELE, VISFATIN and PDGF pathways, leading to ONFH pathology.

There is a lot of transcriptome data here that appear to be competently analysed and that may prove very useful for new studies on all these lineages, cell type-specific regulons and heterotypic cell-cell interactions. However, the biological interpretation is sometimes circular and some of what is presented is incremental to what is well-established; indeed, in this regard, the most novel data are on ECs. My suggestions are to present the data and conclusions in a more straightforward way that acknowledges what is confirmatory or incremental and what is more novel and what new directions could be taken.

Specific comments

1. SC analyses - As noted above, in my opinion the SC data are mainly confirmatory, little new biological ground is broken and the presentation is often circular. First, the fact that there are different stromal cell (SC) types [fibrochondrocytes (FCs), adipocyte lineage cells (ALCs), MSCs, osteoblast lineage cells (OLCs) and chondrocyte lineage cells (CLCs), all at different stages of differentiation or maturity] in the femoral head sampled as the authors have described is well-established. Calling these diverse SC types "SC heterogeneity" is confusing and inaccurate; they represent different lineages. There does however appear to be heterogeneity of individual cells within these different lineages, as also previously described, and this should be more rigorously presented and discussed. Similarly, the fact that the SCs can be positioned into a differentiation sequence with MSCs at origin and branches leading to adipogenesis, osteogenesis and chondrogenesis is also well-established. What the authors might emphasize is what new conclusions and predictions their analyses may allow.

The presentation of how the transcriptional analyses is supported by the histological characteristics of ONFH, HOA and FNF and the different differentiation competence of the SCs from ONFH, HOA and FNF is circular at best and misinterpreted at worst. Since the histological characteristics of ONFH, HOA and FNF are well-established and the authors state they randomly selected regions of the femoral head to collect cells, one predicts and expects the differences seen in frequency of cells within each SC lineage. Similarly, since no fractionation of isolated SCs or MSCs was done prior to differentiation assays, it would seem that one isn't measuring and can't draw conclusions around "differentiation competence" of SCs from tissue of the different conditions. Rather it measures the frequency of different precursor or differentiating cell types of each SC lineage as reflected in the femoral head tissue make-up at isolation. Taken together, these are at best supplemental data to support the bioinformatics of the RNA-seq results.

2. EC analyses - While microvascular dysfunction is a well-established pathological feature in alcohol-induced ONFH, the identification and further transcriptional characterization of ACKR1+ ECs and ACKR1- ECs in the three conditions and the suggestion that the ACKR1+ ECs might affect MSC differentiation via the SELE, VISFATIN and PDGF pathways are more novel and should be the more central part of this manuscript. That being said, Fig. 6 on communication is presented again in a somewhat circular fashion - given that there are different cell populations present, one would expect and predict ligand-receptor differences. Presentation should be tightened up around this, as well as a more cautious attribution of causality to the differences and the pathological outcomes.

Other comments

1. Presentation - The Abstract is not very informative and is confusingly written; the writing there is not up to the standards elsewhere in the manuscript, which overall would benefit from editing. The plethora of abbreviations (which I have succumbed to above for the convenience of the authors) makes the reading of the manuscript difficult; many are unnecessary. The term "regional cells" is meaningless in the context of random collection of tissues from the femoral head, which is a better way of describing what was collected and analysed. Similarly, as above, the term "SC heterogeneity" is not helpful or biologically accurate. Figures - I suggest re-orienting Fig. 2C (and similarly in Fig. 3) to better align with what is shown with Figs. 2A,B, that is where MSCs, ALCs, OLCs, and CLCs are presented relative to the other figure presentations.

2. Description of data. In several places, the data are not presented clearly or as rigorously as

they should be. For example, Fig. 2B - as one would expect based on what is already known, some of the genes aligned under FCs are high in OLCs and ALCs (e.g., Col1A1, FN1) and IBSP aligned under OLCs is high in FCs, again as expected and clearly shown in Fig. 2C. This is an example, but the authors should consider how data are presented throughout.

Point-by-Point Response to Referees

Responses to Reviewer 1

We would like to thank the reviewer for these expert comments on our manuscript, which has been carefully revised according to the reviewer's comments.

This is a very excellent study with the state-of-the-art scRNA-seq technology and fruitful statistical analyses of human stromal and endothelial cells in alcohol-induced ONFH. Liao et al. identified the heterogeneity of stromal and endothelial cells and revealed the differences of their transcriptional information between the case and control groups. In addition, they further illustrated the potential regulatory pathways related to alcohol-induced ONFH through the cell-cell interaction analyses.

1. In line 233, the author demonstrated that “communication between ECs and SCs may play a role in SC differentiation in alcohol-induced ONFH” only based on the cell-cell interaction analyses in alcohol-induced ONFH group. It would be more convincing with the comparison of the cell-cell communications of ECs and SCs between case and control groups.

We agree with the reviewer's comment and have added an intergroup analysis of cell-cell communication. The revised figures and paragraph are shown below.

Figure 6

Figure 6: Analysis of EC and SC communication indicated potential regulatory signaling pathways in alcohol-induced ONFH. A) Heatmap showing the total counts of significant ligand-receptor pairs across all clusters in the ONFH group. The red frame indicates that EC, SC and myeloid cells harbored high-rank numbers of ligand-receptor pairs. **B)** Bubble plot showing specific interaction pathways among ECs and SCs. Pathways in the red frame were markedly different between ACKR1+ ECs and ACKR1- ECs. The bubble size represents the communication probability calculated by CellChat software. Pathway colour represents communication mode (secreted signal or cell-cell contact). **C)** Bar plot showing the communication weights and distribution of ligands and receptors from pathways of interest. The colors in the bar column represent receiving cell groups. **D)** Bar plot showing the proportions of communication pathways in different groups. The upper panel represents the secreted signaling pathway, and the lower panel represents the cell-cell contact pathway. The colours in the bar column represent communication pathways. **E)** Pearson correlation analysis between pathway receptors of interest and adipogenesis or

osteogenesis genes. Pearson-test, * $P < 0.05$

Figure S8

Figure S8: Analysis of communication ligand-receptor counts in the HOA and FNF groups. A-B) Heatmap showing the total counts of significant ligand-receptor pairs across all clusters in the HOA and FNF groups. The red frame and blue frame indicate that EC, SC and myeloid cells harbored high-rank numbers of ligand-receptor pairs.

Figure S9E

E) Pearson correlation analysis between pathway receptors of interest and adipogenesis or osteogenesis genes in SCs among the three groups.

Pearson-test, * $P < 0.05$.

(Results section, line 256) Next, we compared the ECs-SCs communication patterns between groups (Figure 6D). Overall, the cell communication patterns of ACKR1+ ECs and ACKR1- ECs to SCs were different in each group. The VISFATIN (secreted) and SELE (cell-cell contact) pathways were the main bridges of ACKR1+ ECs - SCs communication in the ONFH group. In the HOA group, the MK, MIF (secreted), APP and SELE (cell-cell contact) pathways accounted for a high proportion. In the FNF group, there was an elevated ANGPTL (secreted) pathway.

To further forecast the genetic functions of these differentially expressed pathways, Pearson's correlation analysis was performed between these receptors and classical osteogenic, adipogenic and chondrogenic marker genes in stromal cells (Figure 6E and Figure S9E). In the ONFH group, receptor genes *ITGA5*, *ITGB1* and *CD44* were negatively correlated with osteogenic and adipogenic marker genes but positively correlated with chondrogenic marker genes. The receptor genes *INSR* and *GLG1* showed no tendency during the statistical analysis. In the HOA group, we also found a similar tendency but with a weaker correlation coefficient

in SCs (Figure S9E). The above data suggested that the SELE and VISFATIN pathways had an effect on SC differentiation.

(Discussion section, line 337) Our data suggested that ACKR1+ECs might communicate with CLCs, OLCs and FCs via the VISFATIN pathway through the NAMPT-INSR/NAMPT-ITGA5-ITGB1 axis. In addition, we found that the communication weights of the VISFATIN pathway were upregulated in the ONFH group compared with the other 2 groups. Correlation analysis further revealed that ITGA5 and ITGB1, the main receptors of the VISFATIN pathway, were negatively correlated with osteogenic marker genes and positively correlated with chondrogenic marker genes. These findings provide a new target for the study of abnormal MSC differentiation in ONFH.

SELE (Selectin E), another ACKR1+ ECs-SCs communication pathway associated with stem cell differentiation and elevated in the ONFH group, has been reported to serve as a component of the vascular niche that regulates hematopoietic stem cell dormancy and proliferation [45-47]. Since the SELE pathway is a cell-cell contact type, a prerequisite for this pathway to truly work is vascular injury, so ACKR1+ ECs have the opportunity to contact MSCs in the niche. Immunofluorescence staining showed that ACKR1+ ECs were widely distributed in sinusoids and venules in the ONFH group, making it possible for ACKR1+ ECs to contact MSCs during vascular injury caused by alcohol abuse and other pathogenic factors.

Responses to Reviewer 2

We would like to thank the reviewer for the expert comments on our manuscript, which has been carefully revised according to the reviewer's comments.

This paper by Liao et al. reports single cell RNA-seq results obtained from bone specimens of patients with osteonecrosis of the femoral head (ONFH). The authors describe the differences between the single cell bone transcriptomes of patients with ONFH and hip OA or femoral head fracture (as control). One of their primary findings is an enrichment for ACKR1+ endothelial cells in necrotic samples, which the authors suggest could be associated with pro-inflammatory events in the bone marrow micro-environment. Overall, the experiments appear to be carefully conducted and the data rigorously analyzed. I also found the manuscript to be very well-written. I have only a few comments and questions that I am hoping the authors will address

1. The data presented in Figure 3c and 3d is seemingly contradictory: While ONFH samples are enriched for transcripts associated with adipogenesis, HOA and FNF specimens have more adipocytes. How do the authors interpret this?

We thank the reviewer for this helpful comment. In fact, we also found abundant smaller vacuolar cells in the H&E staining images from the ONFH group, but these cells were not as typical and large as the adipocytes in the HOA and FNF groups. In the previous version, we considered these small vacuolar cells to be immature adipocytes, so we believe that the trends in Figure 3C and in 3D are consistent. However, it was difficult to confirm that the vacuolar cells in the ONFH group were indeed adipocytes based on only H&E staining images. Therefore, we selected chemerin, a known adipokine that was specifically expressed in the ALCs in our sequencing data (Figure S5A), as a marker for IHC detection. The newly added high magnification H&E images and IHC staining pictures (Figure S5B) showed that the bone marrow of the ONFH group was filled with fibrous materials and a large number of small

vacuolar cells with high expression of chemerin, which indicated an identity of adipocyte lineage cells.

Figure S5

Figure S5: Identification of vacuolar cells in bone marrow of the ONFH group. A) Violin plot showing the expression of adipocyte related genes in each cluster of the ONFH group. **B)** Representative bone marrow H&E staining images (left column) and chemerin IHC staining images (right column) for each group, scale bar = 50 µm.

(Result section, line137) H&E staining showed that the bone marrow in the HOA and FNF groups was full of adipose tissue, in which adipocytes were mature and had large lipid droplets (Figure 3C). Conversely, the bone marrow of the ONFH group was filled with fibrous materials and a large number of small vacuolar cells (Figure 3C). To identify these small vacuolar cells, we selected chemerin, a known adipokine [20] specifically expressed in the ALCs in our sequencing data (Figure S5A), as a marker for immunohistochemistry

detection. High levels of chemerin-positive signals were found around these cells (Figure S5B), indicating that these cells were adipocyte lineage cells.

Table S6

Supplementary Table S6. Antibodies information

Reagent name	Source	Identifier
Rabbit monoclonal anti-ACKR1 (EPR5205)	Abcam	ab137044
Mouse monoclonal anti-vWF (F8/86)	Invitrogen	MA5-14029
Rabbit polyclonal anti-VE Cadherin	Abcam	ab33168
Rabbit polyclonal anti-THY1 (JF10-09)	Invitrogen	MA5-32559
Rabbit polyclonal anti-Chemerin	Invitrogen	PA5-77080
PE anti-human CD105 (SN6h)	Biolegend	800503
PE anti-human CD90 (5E10)	Biolegend	328109
PE anti-human CD73 (AD2)	Biolegend	344003
Brilliant Violet 421™ anti-human CD45 (HI3)	Biolegend	304031
PE anti-human CD31 (WM59)	Biolegend	303105
PE anti-human CD14 (63D3)	Biolegend	367103
Goat Anti-rabbit IgG H&L/Cy3 antibody	Bioss	bs-0295G-Cy3
Goat Anti-Mouse IgG H&L/FITC antibody	Bioss	bs-0296G-FITC

2. The methodology of MSC purification and culture is a bit unclear: Were these cells sorted with respect to the aforementioned markers (CD105+CD90+CD73+CD45-CD31-CD14-) and then cultured?

We used the classical continuous passage method to purify MSCs [48,49], and flow cytometry was only used for identification. To purify the cells as much as possible, the cells after the third passage were observed under a microscope, and batches with unified morphology were used for subsequent identification and experiments. We have revised the description in the Method section for clarity.

(Method section, line 460) MSC purification was achieved by continuous passage in plastic culture plates [48,49]. In short, when the primary adherent cells grew to 70-80% confluence, they were digested with 0.25% trypsin, resuspended in new medium and seeded into a new culture plate at a density of 2000/cm². After 2-3 generations of continuous passage according to the above method, the cell morphology was unified into a spindle shape. Cells at passages 3-4 were used for subsequent identification and experiments.

3. Have you checked whether these markers are sufficiently specific to the MSC clusters you identified with scRNA-seq?

We thank the reviewer for this helpful comment. First, we must admit that the MSC cluster in the single-cell sequencing analysis could not correspond to MSCs in vitro through the classical standard (CD105+CD90+CD73+CD45-CD31-CD14-). We attempted to screen all groups of stromal cells based on the above markers, but the results showed that only 2.282% of the total stromal cells fully met the above criteria, and these cells were mainly derived from CLCs and expressed differentiation-related genes (shown below). Obviously, the cells screened by this standard did not correspond to MSCs in vitro. Therefore, we believe that classical surface markers may not be a good standard to define MSCs in scRNA-seq data due to the following. First, the expression of protein markers is not always consistent with the expression of mRNA. Second, the standards (CD105+CD90+CD73+CD45-CD31-CD14-) were initially recommended to identify MSCs cultured in vitro rather than freshly isolated cells [48,49].

Since it is difficult to use the expression of conventional surface markers to determine that the “MSCs” cluster in sequencing data corresponded to MSCs in vitro, we believe that the

annotation of “MSCs” should be modified to avoid misunderstanding. Cluster 21 of the ONFH group, which we annotated as “MSCs”, and SingleR annotated as "Tissue Stem Cells" in previous version did not express differentiation-related markers and was located at the origin of the pseudotime map. All the above clues are indicative of a stem cell signature, so we have redefined these cells as uncommitted stromal cells (USCs).

Although we cannot equate USCs with MSCs, as these two populations may be intersections or subsets, we believe that osteogenic-adipogenic differentiation assays of MSCs in vitro confirmed to some extent the significant differences in stromal cell differentiation in the data analysis. Therefore, we retained the results of the MSC differentiation experiment as supplemental data to support the bioinformatics results. Instead, we have added the results of the DEG and regulon analysis of USCs in Figure 3 to provide more valuable information.

4.The stained bone marrow images are difficult to interpret at the current resolution. Could you perhaps provide higher res images and/or individual channels next to each other so that the overlap (or lack of) between vWF and ACKR1 is more obvious.

We thank the reviewer for this helpful suggestion. We have replaced the image with higher resolution and individual channels to better show vWF and ACKR1. The revised figures and paragraph are shown below.

Figure 5D and E

D and **E**) Representative images of the immunofluorescence staining of ACKR1 and vWF in the bone marrow region of femoral head samples (scale bar = 100 μm) and in the cultured endothelial cells (scale bar = 200 μm). Green fluorescence indicates vWF, red indicates ACKR1, and blue indicates DAPI. Arrows indicate ACKR1+/vWF+ vessels/cells, stars indicate ACKR1-/vWF+ vessels.

(Result section, line 210) ACKR1+/vWF+ cells were mainly observed in sinusoids and venules, but not in arteries (Figure 5D). The number of ACKR1+ vessels and ACKR1+ primary endothelial cells in the ONFH 4 group was greater than that in the other groups (Figure 5D and E), which was consistent with the number of ACKR1+ ECs in the single-cell sequencing data.

5. Further, the appearance of ONFH specimens seem to suggest that arteriole-type vessels were labeled – is this correct? Some comments on this in the main body would be helpful to the reader.

We thank the reviewer for this helpful suggestion. ACKR1+/vWF+ cells were mainly observed in sinusoids and venules but not in arteries (Figure 5D). We have added a description of the labeled vessel types in the revised manuscript.

(Result section, line 210) ACKR1+/vWF+ cells were mainly observed in sinusoids and venules, but not in arteries (Figure 5D).

Responses to Reviewer 3

We would like to thank the reviewer for these expert comments on our manuscript, which has been carefully revised according to the reviewer's comments.

The authors isolated tissue samples from the femoral head of mid- and advanced-stage alcohol-induced osteonecrosis of the femoral head (ONFH) and age-matched hip osteoarthritis (HOA) and femoral neck fracture (FNF) femoral heads. Cells were isolated from the tissue samples and single-cell RNA sequencing (scRNA-seq) was used to compare their transcriptomes. Based on expression profiles and standard bioinformatics approaches, the authors first conclude that there are different stromal cell (SC) types [fibrochondrocytes (FCs), adipogenic lineage cells (ALCs), MSCs, osteogenic lineage cells (OLCs) and chondrogenic lineage cells (CLCs)] that they positioned into a differentiation sequence with MSCs at origin and branches leading to adipogenesis, osteogenesis and chondrogenesis. Comparing ONFH, HOA and FNF samples led to the conclusion that the proportion of the different SC types is different in the 3 conditions and that SC adipogenic and osteogenic differentiation capacity reflected these proportions. None of this is new or unexpected. What is more interesting are the analyses of ECs in the samples, with different proportions of ACKR1+ ECs and ACKR1- ECs in the three conditions and further analysis suggested that the ACKR1+ ECs, with a more pathological or proinflammatory phenotype, might affect MSC differentiation via the SELE, VISFATIN and PDGF pathways, leading to ONFH pathology.

There is a lot of transcriptome data here that appear to be competently analysed and that may prove very useful for new studies on all these lineages, cell type-specific regulons and heterotypic cell-cell interactions. However, the biological interpretation is sometimes circular and some of what is presented is incremental to what is well-established; indeed, in this regard, the most novel data are on ECs. My suggestions are to present the data and conclusions in a more straightforward way that acknowledges what is confirmatory or incremental and what is more novel and what new directions could be taken.

Thank you for recognizing that our study on ECs is interesting. We appreciate your expert comments on the SC analysis. We have addressed the concerns as follows.

Specific comments:

1. As noted above, in my opinion the SC data are mainly confirmatory, little new biological ground is broken and the presentation is often circular. First, the fact that there are different stromal cell (SC) types fibrochondrocytes (FCs), adipocyte lineage cells (ALCs), MSCs, osteoblast lineage cells (OLCs) and chondrocyte lineage cells (CLCs), all at different stages of differentiation or maturity in the femoral head sampled as the authors have described is well-established. Calling these diverse SC types “SC heterogeneity” is confusing and inaccurate; they represent different lineages. There does however appear to be heterogeneity of individual cells within these different lineages, as also previously described, and this should be more rigorously presented and discussed. Similarly, the fact that the SCs can be positioned into a differentiation sequence with MSCs at origin and branches leading to adipogenesis, osteogenesis and chondrogenesis is also well-established. What the authors might emphasize is what new conclusions and predictions their analyses may allow.

The presentation of how the transcriptional analyses is supported by the histological characteristics of ONFH, HOA and FNF and the different differentiation competence of the SCs from ONFH, HOA and FNF is circular at best and misinterpreted at worst. Since the histological characteristics of ONFH, HOA and FNF are well-established and the authors state they randomly selected regions of the femoral head to collect cells, one predicts and expects the differences seen in frequency of cells within each SC lineage. Similarly, since no fractionation of isolated SCs or MSCs was done prior to differentiation assays, it would seem that one isn't measuring and can't draw conclusions around “differentiation competence” of SCs from tissue of the different conditions. Rather it measures the frequency of different precursor or differentiating cell types of each SC lineage as reflected in the femoral head tissue make-up at isolation. Taken together, these are at best supplemental data to support the bioinformatics of the RNA-seq results.

1. We thank the reviewer for these helpful suggestions. Stromal cell differentiation disorder is

one of the most important pathogenic hypotheses during ONFH progression. It is of great value in promoting further exploration of the transcriptome characteristics of stromal cells of different lineages in ONFH. However, clustering and annotation results of sequencing data must be verified before more valuable differential expressed gene analysis and regulon analysis can be conducted. Therefore, we first presented the expression of known lineage markers, the pseudotime map and the GO enrichment results in each stromal cell cluster to strictly verify the annotations to improve the reliability of subsequent analysis. In the revised manuscript, we have added an intergroup comparison of stromal cells and provided more highly variable genes and regulons (Figure 3E and F). The revised figures and paragraph are shown below.

Figure 3

(Result section, line 158) To further explore the underlying molecular mechanisms, we regarded the cells with osteogenic- or adipogenic- differentiation potential (USCs, ALCs and OLCs) as a collection and compared DEGs and regulon activity between groups (Figure 3E, F). NOTCH3, MMP13, MEF2C (27g), ZNF282 (12g), and SMACB1_extend (11g) were found to show higher expression levels or regulon activity in the ONFH 3A group. The activity of transcription factors known to promote osteogenic differentiation, such as JUND_extend (54g), SOX6_extend (17g) and PML_extend (14g), was lower in the ONFH group (Figure 3F).

2. We agree that the statement “SC heterogeneity” is inaccurate, and we have deleted this description in the revised manuscript.

3. In the previous version, we misused the word “randomly” to describe the cell sampling methods. In fact, as illustrated in the diagram in Figure 1A, we sampled bone tissue using a crescent bone-chisel for evenly spaced sampling, and the scope of sampling almost encompassed the coronal plane of the femoral head (new Figure S1). Cells for single-cell RNA sequencing and for differentiation experiments were digested from evenly obtained bone tissue. However, as suggested by the reviewer, we cannot equate the MSCs *in vitro* with any SC cluster or any combination of SC clusters in sequencing data because the MSCs were not sorted before amplification. To avoid misleading the readers, the annotation of “MSCs” in the previous version of bioinformatics result was changed to uncommitted stromal cells (USCs). For the differentiation assay, we believe that osteogenic-adipogenic differentiation assays of MSCs *in vitro* confirm to some extent the significant differences in stromal cell differentiation in the data analysis. Therefore, we retained the results of the MSC differentiation experiment as supplemental data (Figure S7D, E) to support the bioinformatics of the RNA-seq.

Figure 1A

Figure S1

A Coronal plane of the femoral head samples :

B Sampling procedure :

Figure. S1: Sampling procedure of bone tissue acquisition from femoral head samples. A) Coronal plane of the femoral head samples from different groups. B) Specific sampling procedure of bone tissue acquisition. i. Cut the femoral head sample along the coronal plane with a wire saw. ii. Collect the bone tissues from the entire coronal plane with a crescent chisel in 0.5 cm gap. iii. Digest the bone tissue in a mixture of 0.2% (w/v) collagenase type II / 0.2% (w/v) dispase

specific comments 2: EC analyses – While microvascular dysfunction is a well-established pathological feature in alcohol-induced ONFH, the identification and further transcriptional characterization of ACKR1+ ECs and ACKR1- ECs in the three conditions and the suggestion that the ACKR1+ ECs might affect MSC differentiation via the SELE, VISFATIN and PDGF pathways are more novel and should be the more central part of this manuscript. That being said, Fig. 6 on communication is presented again in a somewhat circular fashion – given that there are different cell populations present, one would expect and predict ligand-receptor differences. Presentation should be tightened up around this, as well as a more cautious attribution of causality to the differences and the pathological outcomes.

We thank the reviewer for these helpful suggestions. We have added an intergroup analysis of cell-cell communications. The revised figures and text are shown below.

Figure 6

D) Bar plot showing the proportions of communication pathways in different groups. The upper panel represents the secreted signaling pathway, and the lower panel represents the cell-cell contact pathway. The colours in the bar column represent communication pathways. **E)** Pearson correlation analysis between pathway receptors of interest and adipogenesis or osteogenesis genes. Pearson-test, * $P < 0.05$.

Figure S8

Figure S8: Analysis of communication ligand-receptor counts in the HOA and FNF groups. A-B) Heatmap showing the total counts of significant ligand-receptor pairs across all clusters in the HOA and FNF groups. The red frame and blue frame indicate that EC, SC and myeloid cells harbored high-rank numbers of ligand-receptor pairs.

Figure S9E

E) Pearson correlation analysis between pathway receptors of interest and adipogenesis or osteogenesis genes in SCs among the three groups. Pearson-test, * $P < 0.05$.

(Results section, line 256) Next, we compared the ECs-SCs communication patterns between groups (Figure 6D). Overall, the cell communication patterns of ACKR1+ ECs and ACKR1- ECs to SCs were different in each group. The VISFATIN (secreted) and SELE (cell-cell contact) pathways were the main bridges of ACKR1+ ECs - SCs communication in the ONFH group. In the HOA group, the MK, MIF (secreted), APP and SELE (cell-cell contact) pathways accounted for a high proportion. In the FNF group, there was an elevated ANGPTL (secreted) pathway.

To further forecast the genetic functions of these differentially expressed pathways, Pearson's correlation analysis was performed between these receptors and classical osteogenic, adipogenic and chondrogenic marker genes in stromal cells (Figure 6E and Figure S9E). In the ONFH group, receptor genes *ITGA5*, *ITGB1* and *CD44* were negatively correlated with osteogenic and adipogenic marker genes but positively correlated with chondrogenic marker genes. The receptor genes *INSR* and *GLG1* showed no tendency during the statistical analysis. In the HOA group, we also found a similar tendency but with a weaker correlation coefficient in SCs (Figure S9E). The above data suggested that the SELE and VISFATIN pathways had

an effect on SC differentiation.

(Discussion section, line 337) Our data suggested that ACKR1+ECs might communicate with CLCs, OLCs and FCs via the VISFATIN pathway through the NAMPT-INSR/NAMPT-ITGA5-ITGB1 axis. In addition, we found that the communication weights of the VISFATIN pathway were upregulated in the ONFH group compared with the other 2 groups. Correlation analysis further revealed that ITGA5 and ITGB1, the main receptors of the VISFATIN pathway, were negatively correlated with osteogenic marker genes and positively correlated with chondrogenic marker genes. These findings provide a new target for the study of abnormal MSC differentiation in ONFH.

SELE (Selectin E), another ACKR1+ ECs-SCs communication pathway associated with stem cell differentiation and elevated in the ONFH group, has been reported to serve as a component of the vascular niche that regulates hematopoietic stem cell dormancy and proliferation [45-47]. Since the SELE pathway is a cell-cell contact type, a prerequisite for this pathway to truly work is vascular injury, so ACKR1+ ECs have the opportunity to contact MSCs in the niche. Immunofluorescence staining showed that ACKR1+ ECs were widely distributed in sinusoids and venules in the ONFH group, making it possible for ACKR1+ ECs to contact MSCs during vascular injury caused by alcohol abuse and other pathogenic factors.

Other comments 1: Presentation

1) The Abstract is not very informative and is confusingly written; the writing there is not up to the standards elsewhere in the manuscript, which overall would benefit from editing.

We have revised the Abstract to reduce abbreviations and increase the amount of information.

Alcohol-induced osteonecrosis of the femoral head (ONFH) is a disabling disease with a high incidence and elusive pathogenesis. Here, we used single-cell RNA sequencing to explore the transcriptomic landscape of mid- and advanced-stage alcohol-induced ONFH. Cells derived from age-matched hip osteoarthritis and femoral neck fracture samples were used as control.

Our bioinformatics analysis revealed the disorder of osteogenic-adipogenic differentiation of stromal cells in ONFH and altered regulons such as MEF2C and JUND. In addition, we reported that one of the endothelial cell clusters with *ACKR1* expression exhibited strong chemotaxis and a weak angiogenic ability and expanded with disease progression. Furthermore, ligand-receptor-based cell-cell interaction analysis indicated that *ACKR1*+ endothelial cells might specifically communicate with stromal cells through the *VISFATIN* and *SELE* pathways, thus influencing stromal cell differentiation in ONFH. Overall, our data revealed single cell transcriptome characteristics in alcohol-induced ONFH, which may contribute to the further investigation of ONFH pathogenesis.

2)The plethora of abbreviations (which I have succumbed to above for the convenience of the authors) makes the reading of the manuscript difficult; many are unnecessary.

We have scrutinized our manuscript and deleted unnecessary abbreviations.

3) The term “regional cells” is meaningless in the context of random collection of tissues from the femoral head, which is a better way of describing what was collected and analysed. Similarly, as above, the term “SC heterogeneity” is not helpful or biologically accurate.

We agreed that the term “regional cells” and “SC heterogeneity” were inaccurate, and we have deleted these descriptions in the revised manuscript. We have added Figure S1 (as shown above) to more clearly show our sampling methods and scope.

4) Figures - I suggest re-orienting Fig. 2C (and similarly in Fig. 3) to better align with what is shown with Figs. 2A,B, that is where MSCs, ALCs, OLCs, and CLCs are presented relative to the other figure presentations.

Fig. 2A, B and Fig. 2C were created by using the Seurat package and Monocle2 package, respectively. They technically used different gene sets and algorithms in dimensionality

reduction, so dots representing the same cells were projected to different positions visually. We could hardly overcome this technical obstacle, so we used the same colour to represent the same clusters in Fig.2A, B and C.

Other comments 2: Description of data.

In several places, the data are not presented clearly or as rigorously as they should be. For example, Fig. 2B - as one would expect based on what is already known, some of the genes aligned under FCs are high in OLCs and ALCs (e.g., *Col1A1*, *FN1*) and *IBSP* aligned under OLCs is high in FCs, again as expected and clearly shown in Fig. 2C. This is an example, but the authors should consider how data are presented throughout.

We agree that the description of Figure 2B is unclear and lacks of necessary explanation. In fact, those marker genes were chosen from the DEG list with statistical significance (Table S1), which means that the genes aligned under each cluster were significantly more highly expressed in the corresponding cluster. Some of these genes are indeed expressed in other clusters. In the revised manuscript, we have added a relevant description and explanation, and replaced more specific genes for display to avoid visual misleading. We have also reviewed the manuscript and revised the description with similar problems.

(Result section, line 100) Of note, some cluster feature genes were also expressed at lower levels in other lineages. The osteogenic marker *IBSP* was also expressed in CLCs, the adipogenic marker *APOE* in USCs and OLCs, the stemness feature *NOTCH3* in ALCs and FCs, and feature genes of FCs *COL1A1* and *COL3A1* also in ALCs and CLCs, indicative of the multiple functions of these genes and the cell heterogeneity persisting in the cluster.

(Result section, line 111) Although most of the cells in each cluster had a tendency to concentrate towards a pseudotime node, some of the cells were also scattered in the prebranch.

(Result section, line 115) Moreover, DEG analysis of the two trajectory nodes showed that the top 10 DEGs contained recognized lineage-specific genes (*ADIRF*, *IBSP*, *SPP1*) with expression trends that were consistent with the cell fates (Figure 2E). Some genes (*TPM2*, *SPARCL1*, *OLFML3*, *PTGDS*, *EFEMP1*, *HAPLN1*) lacked a reported connection with MSC differentiation also showed different expression levels among each cell fate.

(Result section, line 137) H&E staining showed that the bone marrow in the HOA and FNF groups was full of adipose tissue, in which adipocytes were mature and had large lipid droplets (Figure 3C). Conversely, the bone marrow of the ONFH group was filled with fibrous materials and a large number of small vacuolar cells (Figure 3C). To identify these small vacuolar cells, we selected chemerin, a known adipokine [20] specifically expressed in the ALCs in our sequencing data (Figure S5A), as a marker for immunohistochemistry detection. High levels of chemerin-positive signals were found around these cells (Figure S5B), indicating that these cells were adipocyte lineage cells.

(Discussion section, line 321) These results suggested that ACKR1+ EC amplification was associated with ONFH disease progression, but it was not clear whether there was a direct causal relationship between the two, which will be revealed by further studies, such as cell-tracing experiments. Another concern was also a certain amount of ACKR1+ ECs in the HOA group, although the amount and proportion were smaller than those in the ONFH 3A group. The role of ACKR1+ ECs in aseptic inflammation and bone remodeling of subchondral bone in osteoarthritis merits further investigation.

REVIEWERS' COMMENTS:

Reviewer #1 (Remarks to the Author):

The authors have addressed my comments satisfactorily and they added several comprehensive analyses to reveal the divergence of the cell communication patterns in each group. This is an excellent study and I don't have any further comments.

Reviewer #2 (Remarks to the Author):

The authors have addressed all of my comments.

Reviewer #3 (Remarks to the Author):

The revisions are well done and have strengthened the presentation and conclusions. I have no remaining concerns.